# Breakdown of the velocity and turbulence in the wake of a wind turbine - Part 2: Analytical modeling.

Erwan Jézéquel[1,2], Frédéric Blondel[1], and Valéry Masson[2]

[1]IFP Energies nouvelles, 1-4 Avenue de Bois Préau, Rueil-Malmaison, France
[2]Centre National de Recherches Météorologiques, 42 avenue Gaspard Coriolis, Toulouse, France

**Correspondence:** Erwan Jézéquel (erwan.jezequel@ifpen.fr)

**Abstract.** This work aims to develop an analytical model for the streamwise velocity and turbulence in the wake of a wind turbine where the expansion and the meandering are taken into account independently. The velocity and turbulence breakdown equations presented in the companion paper are simplified and resolved analytically, using shape functions chosen in the moving frame of reference. This methodology allows us to propose a physically-based model for the added turbulence and thus to have a better interpretation of the physical phenomena at stake, in particular when it comes to wakes in a non-neutral atmosphere. Five input parameters are used: the widths (in vertical and horizontal directions) of the non-meandering wake, the standard deviation of wake meandering (in both directions) and a modified mixing length. Two calibrations for these parameters are proposed: one if the user has access to velocity time series, and the other if he or she does not. The results are tested on a neutral and an unstable LES simulations that were computed with Meso-NH. The model shows good results for the streamwise velocity in both directions and can accurately predict modifications due to atmospheric unstability. For the axial turbulence, the model misses the maximum turbulence at the top tip in the neutral case and the proposed calibrations lead to an overestimation in the unstable case. However, the model shows encouraging behaviour as it can predict a modification of the shape function (from bimodal to unimodal) as unstability, and thus meandering, increases.

## 1 Introduction

The CPU cost of classical computational fluid dynamic models is too high to deal with all the different cases needed to estimate and optimise the performances of a wind farm. Thus, so-called engineering models have been developed to estimate the power loss due to wakes at a low computational cost, e.g. Jensen (1983); Larsen et al. (2008); Bastankhah and Porté-Agel (2014). These design tools are based on physical considerations and are often calibrated and validated against numerical results or measurements. Among these tools, analytical models are the quickest: they consist of a single formula that can be directly applied to the wind farm setup and atmospheric conditions, leading to fast results even for a whole farm. A very commonly used model is the one developed by Bastankhah and Porté-Agel (2014) who assumed an axisymmetric and self-similar Gaussian velocity deficit in the wake and solved the mass and momentum conservation equations to find a relation between the amplitude and width of the Gaussian. It can be adapted for a non-axisymmetric wake (Xie and Archer, 2014):

$$\Delta U(x,y,z) = \frac{\overline{U}_\infty - \overline{U}}{\overline{U}_\infty} = C(x)\exp\left(-\frac{y^2}{2\sigma_y(x)^2} - \frac{z^2}{2\sigma_z(x)^2}\right) \tag{1}$$

$$C(x) = 1 - \sqrt{1 - \frac{C_T}{8\sigma_y(x)\sigma_z(x)/D^2}} \tag{2}$$

where $\overline{U}$ is the mean velocity field, $\overline{U}_\infty$ is the mean velocity upstream of the turbine, $C(x)$ is the maximum velocity deficit, $C_T$ is the thrust coefficient, $D$ is the turbine diameter, $(x,y,z)$ are the streamwise, lateral and vertical coordinates, centred at the turbine's hub, and $\sigma_{y,z}$ the wake widths in the lateral and vertical directions. In the present work, the vertical and horizontal axes are centred at the hub position. Here and in the following, the Reynolds decomposition is used to write any unsteady field $X(t)$ as a sum of a mean and a varying part: $X(t) = \overline{X} + X'(t)$.

The stability of the atmospheric boundary layer (ABL) influences the wake recovery (Abkar and Porté-Agel, 2015) and the large-scale eddies carried in this region of the atmosphere are often associated with wake meandering, i.e. oscillations of the instantaneous wake around its mean position (Larsen et al., 2008). To model the meandering, the concepts of fixed and moving frames of reference (respectively denoted FFOR and MFOR) defined in the dynamic wake meandering (DWM) model are used herein (Larsen et al., 2007). The FFOR is bound to the ground: it is the frame of reference in which we want to compute the turbulence and velocity fields. In the FFOR the effects of meandering are not differentiated from the wake expansion caused by turbulent mixing. The MFOR is moving with the wake centre at each time step: in this frame of reference, only the wake expansion due to turbulent mixing is represented, making the fields in this frame of reference easier to interpret. The instantaneous streamwise velocity can be changed from one frame to another according to the relation:

$$U_{MF}(x,y,z,t) = U_{FF}(x, y + y_c(x,t), z + z_c(x,t), t) \tag{3}$$

where subscripts MF and FF denote the velocity fields in the MFOR and FFOR respectively, $y_c(x,t)$ and $z_c(x,t)$ are the time series of the wake centre's coordinates at the downstream position $x$. The concept of MFOR and FFOR can be used to write an analytical wake model for the velocity deficit as in the work of Braunbehrens and Segalini (2019):

$$\Delta U_{FF}(y,z) = C\left[1 + \left(\frac{\sigma_{fy}}{\sigma_y}\right)\right]^{-1/2}\left[1 + \left(\frac{\sigma_{fz}}{\sigma_z}\right)\right]^{-1/2}\exp\left[-\frac{y^2}{2\sigma_y^2 + 2\sigma_{fy}^2} - \frac{z^2}{2\sigma_z^2 + 2\sigma_{fz}^2}\right] \tag{4}$$

where $\sigma_{fy,fz}(x)$ are the standard deviations of the wake centre's coordinates in the lateral and vertical directions respectively, $\sigma_{y,z}(x)$ are the wake widths in the MFOR and $C(x)$ is the maximum velocity deficit in the MFOR. Such a model allows calibrating independently the effects of meandering (through the variables $\sigma_{fy,fz}$) and of wake expansion due to turbulent mixing (through the variables $\sigma_{y,z}$). The former parameters are a function of atmospheric stability through lateral and vertical turbulence (Braunbehrens and Segalini, 2019; Du et al., 2021; Brugger et al., 2022) whereas the latter parameters can be a function of axial turbulence as in Eq. 1 (Fuertes et al., 2018; Niayifar and Porté-Agel, 2016) or turbine operating conditions such as $C_T$ and atmospheric shear (Braunbehrens and Segalini, 2019).

For the turbulent kinetic energy (TKE), it is common to model only the maximum value of added turbulence which can be computed with the Crespo model (Crespo and Hernandez, 1996) or the Frandsen model (Frandsen, 2007) as in the IEC 61400-1 standard. Their approach is mainly empirical and can be extended to describe the whole profile of turbulence instead

of the maximum value alone (Ishihara and Qian, 2018). This widely used model (hereafter denoted I&Q2018) is simple since it only requires the knowledge of the thrust coefficient and the upstream turbulence intensity, but it is totally empirical and does not account for atmospheric stability. Moreover, it has been shown that the wake in an unstable ABL dissipates faster than in a neutral ABL even at the same level of turbulence intensity (Du et al., 2021). This behaviour cannot be taken into account in the I&Q2018 model due to the limited number of inputs.

The present work aims to propose a physically-based model that predicts both the mean and variance (i.e. turbulence) of the axial velocity in the wake of a wind turbine. The advantage of basing our model on physical interpretations is that it gives more room for further improvements, as we know which assumptions were made, and how it degrades the results. Moreover, the proposed model is dependent on atmospheric stability, since it influences both the velocity and the turbulence fields in the wake (see companion paper). Many models, such as the I&Q2018 model do not take atmospheric stability into account, assuming

that stable and unstable cases compensate each other and thus a calibration on neutral cases is sufficient. This approach is valid for monthly or yearly estimations of wind farms' performances. But some applications of the future wind industry such as digital twins need estimations over a day, an hour, or even smaller periods. In such cases, the stability must be taken into account. Since we showed in the companion paper that stability mainly affects the wake meandering, this phenomenon must be decoupled from the wake expansion to take the ABL stability into account. To do so, the breakdowns described in the

companion paper are reused and quickly reminded in the following lines.

A field in the MFOR can be written as an unsteady translation of the same field in the FFOR through Eq. 3. To shorten this equation, the notation $\widehat{a(y,z)} = a(y - y_c(t), z - z_c(t))$ for any field $a$, is used. For the present work, it is also important to note that for any field $a$:

$$\overline{\widehat{a}} = \overline{a} ** f_c \tag{5}$$

where $**$ denotes a 2D convolution and $f_c$ is the probability density function (PDF) of the wake centre position. In the companion paper, it has been shown that the velocity (Eq. 6) and turbulence (Eq. 7) in the FFOR can be expressed as a function of their counterparts in the MFOR. This is achieved by decomposing these quantities into several terms, noted (I) and (II) in Eq. 6 and (III) to (VII) in Eq. 7.

$$\overline{U_{FF}} = \underbrace{\overline{\widehat{U_{MF}}}}_{(I)} + \underbrace{\overline{\widehat{U'_{MF}}}}_{(II)} \tag{6}$$

$$k_{FF} = \underbrace{\overline{\widehat{\overline{U_{MF}}}^2} - \overline{\widehat{\overline{U_{MF}}}}^2}_{k_m = (III)} + \underbrace{\overline{\widehat{k_{MF}}}}_{k_a = (IV)} + \underbrace{2\mathrm{cov}\left(\overline{\widehat{U_{MF}}}, \widehat{U'_{MF}}\right)}_{(V)} + \underbrace{\overline{(\widehat{U'^2_{MF}})'}}_{(VI)} - \underbrace{\overline{\widehat{U'_{MF}}}^2}_{(VII)} \tag{7}$$

These terms are thoroughly described and quantified in the companion paper where they are separated into pure-terms ((I),(III) and (IV)) and cross-terms ((II), (V), (VI) and (VII)).

The term (I) is the convolution of $\overline{U_{MF}}$ with $f_c$. It is a pure mean velocity term: it is null only if the mean velocity is null. Conversely, the term (II) is a cross-term because it can be equal to 0 either if there is no meandering (operator $\widehat{\phantom{x}}$ has no effect) or if there is no turbulence in the MFOR ($U'_{MF} = 0$). The term (III), also written $k_m$ in the following to be consistent with notation from Keck et al. (2013) and Conti et al. (2021), is the turbulence purely induced by meandering: in the case of a meandering steady wake i.e. $U'_{MF} = 0$, Eq. 7 reduces to this term only. The term (IV) is the rotor-added turbulence, which is also written $k_a$ for consistency with other works (Conti et al., 2021). It is the turbulence purely induced by the rotor: in absence of meandering, the equation reduces to this term only, also written $k_a$ in the following for consistency with the literature. Term (V) is the covariance of $\widehat{\overline{U_{MF}}}$ and $\widehat{\overline{U'_{MF}}}$, term (VI) can be viewed as the varying part of the MFOR turbulence and term (VII) is the square of the term (II). It is a pure dissipation term as it is always negative. Like the term (II), they are cross-terms since they are equal to zero if either the turbulence in the MFOR or the meandering is null. The companion paper showed that terms (II) and (VII) are negligible in their respective equations. In the breakdown of the turbulence equation, the term (V) is of lesser importance than (III) and (IV) but drives the vertical asymmetry of the turbulence profiles.

The proposed analytical model is based on the velocity and turbulence breakdowns (Eqs. 6 and 7). Similarly to Eq. 4 (Braunbehrens and Segalini, 2019), the reasoning starts by writing the wake properties in the MFOR and the wake meandering with different parameters to take into account meandering due to atmospheric stability independently of the expansion due to turbulence mixing. It is common in wake modelling to assume that meandering can be entirely accounted for by increasing the wake expansion. However, it is a phenomenon of different nature and it leads to velocity and turbulence profiles of different shapes. In the present model, these phenomena are modelled separately, and it will be assumed that they do not interact. This is equivalent to neglecting cross-terms in Eqs. 6 and 7 which have been shown to take consistently smaller values than pure-terms in the companion paper. In the future though, modelling these cross-terms might be necessary to improve the results. The main added value of this work is to propose a new framework that can be used with different shape functions in the MFOR to propose other turbulence models. Nevertheless, two calibrations (one requiring the inflow time series, and another that does not) are proposed for the model, to demonstrate how it can be tuned and to test the model.

In the second section of this work, the datasets are presented: for the calibration of the model, a dataset from the MOMENTA project is used, and for the validation the neutral and unstable cases obtained from the large eddy simulations (LESs) from the companion paper are reused. The third section presents the derivation of the model. The fourth section shows the chosen calibration methods and the fifth section presents the corresponding results. All these results are discussed in a sixth section, followed by the conclusion.

## 2 The LESs datasets

### 2.1 Description of the LES code

The analytical model developed in this work is based on LESs datasets generated with the Meso-NH solver (Lac et al., 2018). It is a finite volume and finite difference research code for ABL simulations where the Navier-Stokes equations and the energy conservation equation are resolved on an Arakawa C-grid. This solver models the stability of the ABL with a buoyancy term in the momentum equation. The Coriolis force and large-scale forcing are also taken into account. The effect of the wind turbine on the surrounding flow is modelled with an actuator line method, i.e. rotating source terms in the momentum equation.

To close the set of equations, the subgrid TKE equation is resolved, and all the subgrid quantities are written as a function of this subgrid TKE, the resolved variables and a Deardorff mixing length. A grid nesting method allows having simultaneously a vertical and horizontal mesh size of $1.5$ m and $0.5$ m in the wake region for the two datasets, and a domain large enough to compute the largest eddies of the atmosphere. The model and numerical parameters are described in more detail in the companion paper.

### 2.2 Simulation setup

Two different LESs datasets are used in this work: the first one for creating and calibrating the model and the second one for testing the model's results. Inflow conditions of these datasets can be found in Table 1. For both datasets, only the wake mean streamwise velocity ($\overline{U}_x$, written $U_x$ in the following), and the streamwise turbulence ($k_x = \overline{u'u'}$) are computed. The proposed model thus only deals with the streamwise velocity and turbulence.

The calibration dataset contains 6 simulations, with four different ABL stabilities and three different thrust values. The simulated turbine is of 92 meters in diameter and hub height of 80 meters. The turbine's data were obtained in the context of the MOMENTA project (Jézéquel, 2023).

To perform such simulations, a precursor without heat flux is first simulated in a domain of 19 km x 15 km (with a horizontal resolution of 37.5m) during 25 hours to let the turbulence establish and the system to reach a quasi-steady-state. Then, a ground heat flux is applied for 4 hours: $0W/m^2$, $30W/m^2$, $60W/m^2$, and $120W/m^2$ for cases 'Neutral', 'Weakly unstable', 'Unstable' and 'Strongly unstable' respectively. This allows to simulate three different levels of atmospheric unstability, starting from the same neutral state. No stable case was simulated because of the induced veer (gradient of inflow wind direction) that leads to a deformed wake. The veer could have been modelled as in Abkar et al. (2018) but it would significantly complicate the present derivations. Moreover, meandering and meandering turbulence are negligible in a stably stratified ABL (see companion paper) and thus there is little interest in using the approach presented herein. Developing the model for veered cases is a challenge that is out of the scope of this work.

After these two steps, the coarsest computational domain (horizontal resolution of 37.5 m) is ready: two grid nestings are then applied to reach a resolution of 1.5 m in the most refined domain. Then, 10 minutes of dynamics are used to let the flow establish in the wake of the wind turbine, and the post-processing is performed on the following 50 minutes of dynamics. The

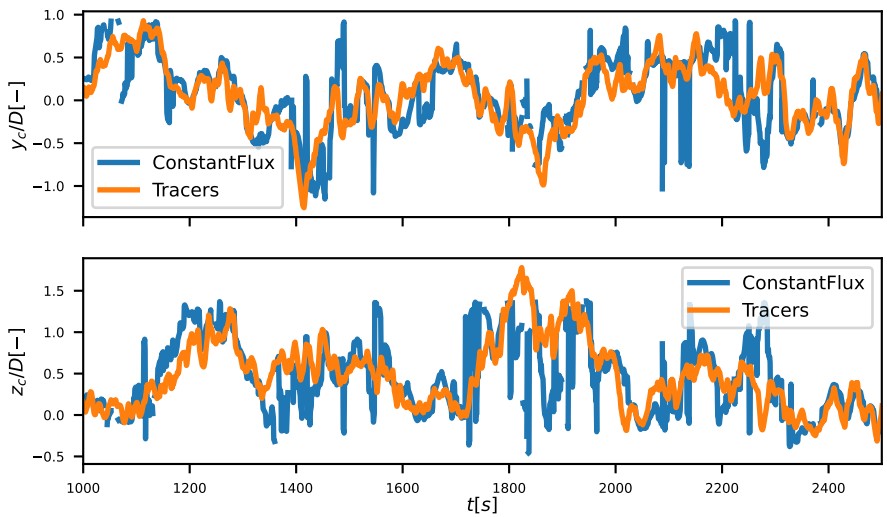

**Figure 1.** Time series of the wake centre's lateral (top) and vertical (bottom) coordinates with the ConstantFlux method and the pollutant method. Weakly unstable case at $x/D = 6$ between 1000 and 2500 seconds.

data is sampled at $1.2$ Hz, which is the approximate limit between resolved and subgrid TKE for these simulations (equivalent to four times the mesh size).

The wind turbine rotational speed and pitch are set according to the controller's database and the calculated upstream velocity. Since all the cases are computed at a similar inflow velocity, similar values of the thrust coefficient are obtained in the simulations. To have the influence of the thrust coefficient on the model, two additional cases with a degraded thrust coefficient are also computed, with the same inflow as the neutral case. To reduce the thrust, the pitch value is increased from 0 to 3 and 4.5 degrees respectively.

The second set of simulations, hereafter called validation dataset, is based on the neutral and unstable cases that are described in the companion paper. The simulated turbine is a modified version of the Vestas V27: it is a three-bladed rotor with a diameter $D = 27$ m and a hub height of $32.1$ m. The simulation methodology is quite similar as described in the paragraph above, except that one additional nesting is required to reach the targeted mesh size. In the validation dataset, the velocity is sampled at 1 Hz and the simulations last for 80 and 40 minutes for the neutral and unstable cases respectively. This was due to
benchmark requirements and computational limitations. A statistical convergence of our datasets is proposed in the appendix of the companion paper. Overall, it concluded that increasing the duration of simulation for the unstable case would improve the reliability of the simulations

### 2.3   Wake tracking

For the validation simulations, the wake centre's coordinates $y_c(x,t)$ and $z_c(x,t)$ are computed at each time step and each
downstream position with the *Constant Flux* wake tracking algorithm, which is described in the companion paper. To facilitate

the wake tracking, a *Reference* simulation is also run. It is a simulation with the same inflow and boundary conditions but without the wind turbine. The corresponding velocity field noted $U_{ref}$ is thus representing a developing ABL without the perturbations of a wind turbine.

Another method is here proposed to compute the unsteady wake centres in the calibration dataset. A passive scalar (similar to a pollutant) is emitted at the rotor disk with a concentration value of 1 at each time step. This new variable is only driven by the advection scheme, in accordance with the passive tracer of the DWM theory, and impairing only marginally the code's performance. By supposing that this variable follows the wake, the unsteady wake centre is deduced from the centre of mass of this pollutant at each downstream position. The results lead to a low-frequency behaviour similar to the ConstantFlux method used in the companion paper but with fewer outliers (see Fig. 1). Since the post-process is more straightforward and the results seem better, this method has been used for the calibration dataset.

## 2.4 Inflow conditions

Table 1 shows the hub height velocity, thrust coefficients and turbulence intensities at hub height for each of the cases. The directional turbulence intensities are defined as:

$$I_{x,y,z} = \frac{\sqrt{k_{x,y,z}}}{U_{\infty,hub}} \tag{8}$$

and the global turbulence intensity is defined as:

$$I = \sqrt{\frac{1}{3}\left(I_x^2 + I_y^2 + I_z^2\right)} \tag{9}$$

| | Name | $U_{\infty,hub}[m\ s^{-1}]$ | $C_T[-]$ | $I[-]$ | $I_x[-]$ | $I_y[-]$ | $I_z[-]$ |
|---|---|---|---|---|---|---|---|
| | Neutral | 7.0 | 0.68 | 0.088 | 0.106 | 0.086 | 0.069 |
| | Weakly unstable | 7.3 | 0.67 | 0.098 | 0.106 | 0.101 | 0.085 |
| Calibration | Unstable | 7.0 | 0.70 | 0.122 | 0.100 | 0.164 | 0.087 |
| | Strongly unstable | 7.0 | 0.70 | 0.153 | 0.154 | 0.179 | 0.112 |
| | Pitch 3 | 7.0 | 0.51 | 0.091 | 0.109 | 0.089 | 0.071 |
| | Pitch 4.5 | 7.0 | 0.43 | 0.092 | 0.115 | 0.086 | 0.072 |
| Validation | Neutral | 8.3 | 0.79 | 0.093 | 0.114 | 0.087 | 0.072 |
| | Unstable | 6.1 | 0.82 | 0.119 | 0.125 | 0.148 | 0.070 |

**Table 1.** List of LES cases

Figures 2 and 3 show the profiles of some inflow variables for the calibration and validation cases, respectively. The profiles are taken 2.5 diameters upstream of the wind turbine and are averaged along the y direction (the direction transverse to the wind turbine).

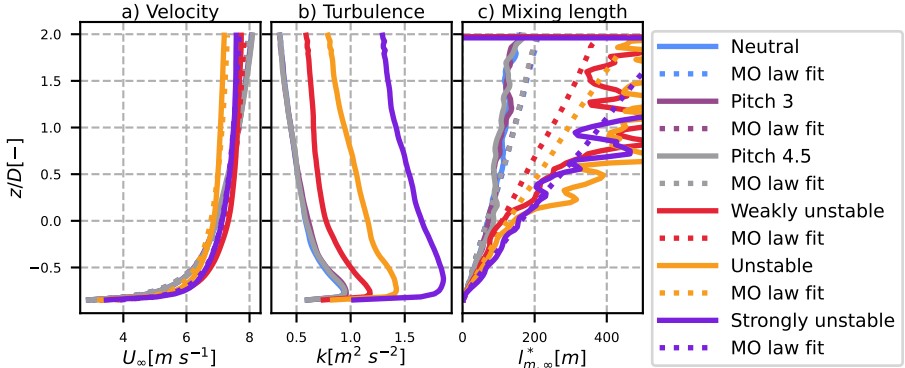

**Figure 2.** Inflow conditions for the calibration cases. a) Mean velocity profile; b) Mean TKE profile; c) Mean $k_x$-to-shear ratio profile. Solid lines: LES results; dotted lines: fit with the Monin-Obukhov law

In the left panel is plotted the mean velocity. The calibration dataset (Fig. 2) has been built in order to have similar hub height velocities between the cases (around 7 m s[-1]) whereas the validation dataset comes from simulations that reproduced the SWiFT benchmark, where the hub height velocities differed. In dotted lines are plotted the Monin-Obukhov profiles:

$$U(z) = \frac{u_*}{\kappa} \left( \ln(z/z_0) + \psi(z, L_{MO}) \right) \tag{10}$$

where $\kappa = 0.41$ is the von Karman constant and (Cheng et al., 2019):

$$\psi(z, L_{MO}) = -2\ln((1+x_u)/2) - \ln((1+x_u^2)/2) + 2\arctan(x_u) - \pi/2 \tag{11}$$

and $x_u = (1 - 15 \cdot z/L_{MO})^{0.25}$. Since $z_0$ is known from the simulations (0.17 in the calibration dataset and 0.014 in the validation dataset), the profiles are found by fitting Eq. 10 on the corresponding velocity profile, with parameters $u_*$ and $L_{MO}$. The results, in dotted lines, match well the inflow profiles, showing that it respects the Monin-Obukhov similarity theory around the turbine's height.

The middle panels of Figs 2 and 3 show the inflow TKE, defined as:

$$k = \frac{1}{2} \left( k_x + k_y + k_z \right) = \frac{3}{2} \left( I \cdot U_{\infty, hub} \right)^2 \tag{12}$$

In the calibration dataset, the amount of TKE increases as the imposed heat flux increases. In the validation dataset, this is not the case since the neutral case is at a higher velocity at hub height, but the TI of the unstable case is indeed higher than that of the neutral case.

The right panels of Figs 2 and 3 show the modified mixing length $l_{m,\infty}^*$ upstream the wind turbine. This quantity will be discussed and used in Sec. 3 to compute the mixing length in the MFOR. Here, the value is computed as the ratio of turbulence and shear:

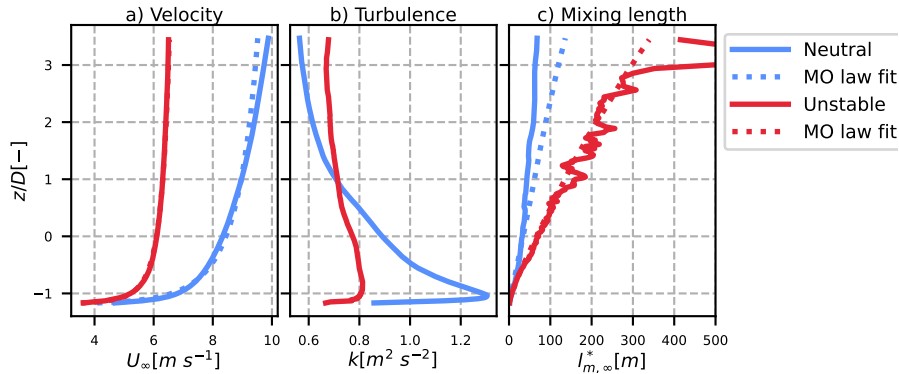

**Figure 3.** Inflow conditions for the validation cases. a) Mean velocity profile; b) Mean TKE profile; c) Mean $k_x$-to-shear ratio profile. Solid lines: LES results; dotted lines: fit with the Monin-Obukhov law

$$l_{m,\infty}^* = \frac{\sqrt{k_{x,\infty}}}{\dfrac{\partial U_\infty}{\partial z}}. \tag{13}$$

However, in the unstable cases, the velocity profile becomes nearly constant above a given height, leading to low values of $\partial U_\infty/\partial z$ and thus very chaotic behaviour of $l_{m,\infty}^*$. To have a more reliable curve, the derivative of $U$ is resolved analytically using Eq. 10:

$$\frac{\partial U_\infty}{\partial z} = \frac{u_*}{\kappa z}\left(1 - 15z/L_{MO}\right)^{-0.25} \tag{14}$$

with $L_{MO}$ and $u_*$ fitted from the velocity profile. The resulting curve, in dotted lines, gives a more useful quantity on the turbulence-to-shear ratio, while still being on the order of magnitude of the directly computed ratio (in solid line).

## 3   Model derivation

In this section, we derive an analytical model for the dominating terms of Eqs. 6 and 7. First, an analytical form is proposed for the velocity deficit in the MFOR $\Delta \overline{U}_{x,MF}$, the turbulence in the MFOR $k_{x,MF}$ and the meandering distribution $f_c$. Then, some terms are neglected and the convolutions of Eqs. 6 and 7 are resolved analytically to get a model for the velocity deficit and added turbulence in the FFOR. To help the reader, the main variable notations and subscripts used in this section and afterwards are summarised in Table. 2.

| $k$ | $k_m$ | $k_a$ | $._x$ | $._{am}$ | $C$ |
|---|---|---|---|---|---|
| Turbulence | Meandering turbulence i.e. term (III) | Rotor-added turbulence i.e. term (IV) | x-component of the vector | Analytical model | Amplitude of the velocity deficit |
| | | | | | |
| $\sigma$ | $\sigma_f$ | $l_m^*$ | $f_c$ | $K_{MF}$ | |
| Velocity deficit width in the MFOR | Variance of the wake centre | Modified mixing length | PDF of the wake centres | Amplitude of the turbulence in the MFOR | |

**Table 2.** Description of the most used notations in this part and the following

## 3.1 Independent modelling of the wake in the MFOR and meandering

### 3.1.1 Wake velocity deficit in the MFOR

Based on the literature (Bastankhah and Porté-Agel, 2014; Xie and Archer, 2014), the mean velocity deficit is modelled with the long-established Gaussian velocity deficit (cf Eq. 1):

$$\Delta U_{x,MF,am}(x,y,z) = C(x)\exp\left(-\frac{y^2}{2\sigma_y^2(x)} - \frac{z^2}{2\sigma_z^2(x)}\right) \tag{15}$$

where subscript $._{am}$ stands for "analytical model", $C(x)$ is defined in Eq. 2 and $\sigma_y, \sigma_z$ are the wake widths in the MFOR. The overline is dropped because this analytical model is static. In the literature, it has been shown that double-Gaussian (Keane et al., 2016) or super-Gaussian (Blondel and Cathelain, 2020) shapes provide more accurate results, but here the Gaussian shape allows a straight-forward computation of the convolutions in our model and is still pertinent in the far wake. It will be shown that this approximation leads to discrepancies in the near wake.

### 3.1.2 Wake added turbulence in the MFOR

To model term (IV) or $k_{x,a}$, one needs an analytical form for the turbulence in the MFOR $k_{x,MF}$. It was first thought better to model the added-turbulence in the MFOR, i.e. $\Delta k_{x,MF} = k_{x,MF} - k_{x,\infty}$, in order to separate the rotor-added turbulence $\Delta k_{x,MF}$ from the ambient turbulence. This procedure was done in the companion paper, however it leads to negative values of $\Delta k_{x,MF}$ (in particular near the ground), i.e. smaller turbulence in the wake compared to the turbulence upstream of the wind turbine. This is not compatible with a model that predicts only increased turbulence in the wake of a turbine (as here or in I&Q2018) and thus this approach has been abandoned.

The derivation of a model for $k_{x,MF}$ is not as straightforward as for $\Delta U_{MF}$ because turbulence comes from the unsteadiness of the flow whereas an analytical model is by definition steady. In the DWM, the Madsen formulation (Madsen et al., 2010) is used to scale the velocity profile with an empirical function of the wake-generated shear. One could also assume self-similarity of the $\Delta k_{x,MF}$ function and try to derive a model as it was done for the velocity in Bastankhah and Porté-Agel (2014). The

main issue here is that an analytical form of the model is needed in the FFOR, i.e. the convolution of $f_{c,am}$ with the chosen shape function for $\Delta k_{x,MF,am}$ must have an analytical solution, which is not trivial for the aforementioned models.

It is here proposed to assume that the turbulence in the MFOR is solely driven by wake-generated shear as in Madsen et al. 235 (2010). To relate the turbulence in the MFOR to mean gradients, two models for the velocity scale $u_0$ are combined. In the first, it is assumed to be proportional to the square root of the TKE (Pope, 2000). However in the present work, the three-dimensional TKE is not computed, so it is replaced with the axial turbulence $k_x$:

$$u_0 = C_\mu^{1/4} k_x^{1/2}. \tag{16}$$

where $C_\mu$ is a constant and $l_m$ is the mixing length. In the second method, the velocity scale is defined from the norm of the 240 strain-rate tensor $|\overline{\overline{S}}|$:

$$
\begin{aligned}
u_0 &= l_m |\overline{\overline{S}}| \\
&= l_m \cdot \sqrt{\left(\frac{\partial U_x}{\partial x}\right)^2 + \left(\frac{\partial U_y}{\partial y}\right)^2 + \left(\frac{\partial U_z}{\partial z}\right)^2 + \frac{1}{2}\left(\frac{\partial U_x}{\partial y} + \frac{\partial U_y}{\partial x}\right)^2 + \frac{1}{2}\left(\frac{\partial U_x}{\partial z} + \frac{\partial U_z}{\partial x}\right)^2 + \frac{1}{2}\left(\frac{\partial U_y}{\partial z} + \frac{\partial U_z}{\partial y}\right)^2}
\end{aligned}
\tag{17}
$$

From the literature (Iungo et al., 2017), it appears that in the wake of a wind turbine, the dominating term (in cylindrical coordinates) is $\frac{\partial U}{\partial r}$. It is supposed herein that these results can be transposed in Cartesian coordinates and are applicable in the 245 MFOR. The velocity scale can thus be written as a function of the derivatives of the axial velocity.

$$u_0 = l_m \cdot \sqrt{\frac{1}{2}\left(\frac{\partial U_x}{\partial y}\right)^2 + \frac{1}{2}\left(\frac{\partial U_x}{\partial z}\right)^2} \tag{18}$$

Combining Eqs. 16 and 18 leads to:

$$
\begin{aligned}
k_{x,MF,am} &= \left(\frac{u_0}{C_\mu^{1/4}}\right)^2 \\
&= \frac{l_m^2}{2C_\mu^{1/2}} \cdot \left[\left(\frac{\partial U_{x,MF}}{\partial y}\right)^2 + \left(\frac{\partial U_{x,MF}}{\partial z}\right)^2\right] \\
&= \frac{l_m^2}{2C_\mu^{1/2}} \cdot \left[\left(-U_\infty(z)\frac{\partial \Delta U_{MF}}{\partial y}\right)^2 + \left(-U_\infty(z)\frac{\partial \Delta U_{MF}}{\partial z} + (1 - \Delta U_{MF})\frac{\partial U_\infty(z)}{\partial z}\right)^2\right]
\end{aligned}
\tag{19}
$$

In Eq. 19, the last term $(1 - \Delta U_\infty)\frac{\partial U_\infty(z)}{\partial z}$ represents the produced turbulence due to the interaction between wake generated shear and atmospheric shear. It is this term that induces a maximum of turbulence at the top tip in cases of high atmospheric shear such as neutral or stable ABLs. Even though an analytical form of this term can be found by assuming $U_\infty(z)$ as a log law or a power law, the convolution product with $f_c$ in Eq. 7 did not lead to any analytical solution.

It was thus decided to neglect shear in the formulation and to add the contribution of the inflow turbulence with a maximum function. This is a strong assumption that impacts the results (see Sect. 5), but allows to compute the total added turbulence:

$$(IV)_{am} = k_{x,a,am} = \max\left(k_{x,\infty}, f_c * * k_{x,MF,am}\right) \tag{20}$$

with:

$$
\begin{aligned}
k_{x,MF,am} &= (U_\infty(z)l_m^*)^2 \left[ \left( \frac{\partial \Delta U_{MF,am}}{\partial y} \right)^2 + \left( \frac{\partial \Delta U_{MF,am}}{\partial z} \right)^2 \right] \\
&= K_{MF}(x,z) \left[ \left( \frac{y}{\sigma_y^2(x)} \right)^2 + \left( \frac{z}{\sigma_z^2(x)} \right)^2 \right] \exp\left( -\frac{y^2}{\sigma_y^2(x)} - \frac{z^2}{\sigma_z^2(x)} \right)
\end{aligned}
\tag{21}
$$

where $K_{MF} = (U_\infty C l_m^*)^2$ and $l_m^*$ is the modified mixing length $l_m^* = l_m / \sqrt{2} C_\mu^{1/4}$. In other words, the modified mixing length is the ratio of the axial turbulence to the quadratic sum of the vertical and horizontal gradients of the axial velocity deficit.

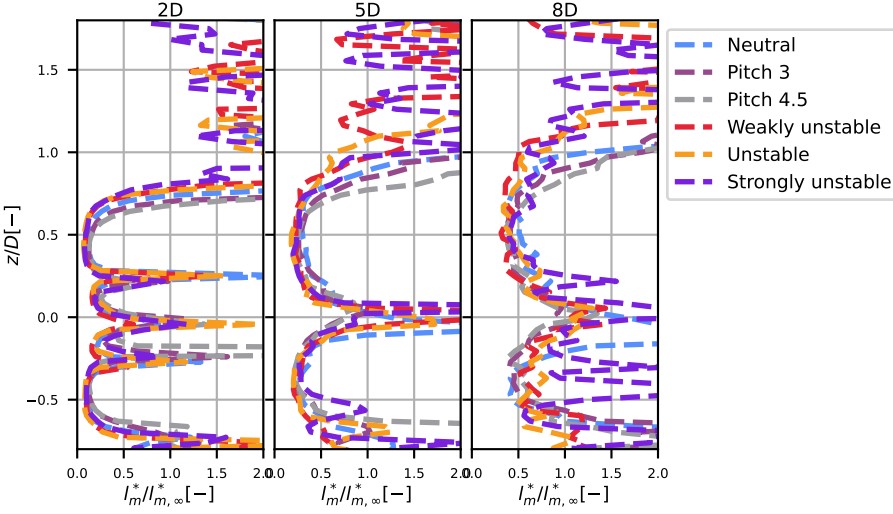

**Figure 4.** Profiles of modified mixing length (turbulence to shear ratio) for the different simulations

Figure 4 shows the profiles of the modified mixing length in the wake normalized by the modified mixing length upstream
the turbine, at hub height: $l_m^*/l_{m,\infty}^*(z_{hub})$ where $l_{m,\infty}^*$ is defined in Eq. 13 and $l_m^*$ is computed as:

$$
l_m^* = \frac{\sqrt{k_{x,MF}}/U_\infty}{\sqrt{\left( \frac{\partial \Delta U_x}{\partial y} \right)^2 + \left( \frac{\partial \Delta U_x}{\partial z} \right)^2}}
\tag{22}
$$

One can see that there are two distinct values: one inside the wake and one outside the wake. Inside the wake, the value is fairly constant (except in the bottom of the wake where it increases chaotically, probably due to the effect of the ground). It only seems to vary with the streamwise distance and thus it was chosen to assume that $l_m^*/l_{m,\infty}^*(z_{hub})$ is only dependent on $x$. Theoretically, it could be possible to develop a model with two mixing lengths (one for the wake and another for the ambient turbulence) but with such an assumption, no analytical solution of Eq. 7 could be achieved.

Note that in Eq. 21, the error in the near-wake due to the Gaussian shape assumption for velocity deficit in the MFOR propagates onto $\Delta k_{x,MF,am}$. Using a Gaussian instead of a super-Gaussian function leads to an underestimation of the wake-generated shear and thus to a much weaker but more spread axial turbulence around the blade's tips. Moreover, the model does not account for the atmospheric-generated shear. This phenomenon, which leads to a smaller value of wake-generated turbulence at the bottom tip compared to the top tip, cannot be represented in our model. Finally, the model imposes that $k_{x,MF,am} = 0$ at the centre of the wake, a condition that is not fulfilled in the calibration dataset. Another possible improvement would be to add the streamwise gradient $\partial U_x/\partial x$ in Eq. 18. Despite these flaws, this expression has been chosen since it has an analytical solution of its convolution with the wake centre position distribution $f_{c,am}$.

### 3.1.3   Wake meandering

For the PDF of wake meandering, the central limit theorem leads to a Gaussian distribution (Braunbehrens and Segalini, 2019). The distribution of the wake centre $f_c$ is non-axisymmetric and thus its variance $\sigma_f$ is defined in both dimensions:

$$f_{c,am}(x,y,z) = \frac{1}{2\pi\sigma_{fy}(x)\sigma_{fz}(x)}\exp\left(-\frac{y^2}{2\sigma_{fy}^2(x)} - \frac{z^2}{2\sigma_{fz}^2(x)}\right) \tag{23}$$

### 3.2   Velocity in the FFOR

In the following, the dependency of the variables on coordinate $x$ is omitted to lighten the equations.

In Eq. 6, the velocity in the wake is written under its dimensional form whereas the model chosen in Eq. 15 is written under the velocity deficit form. To relate the velocity to the velocity deficit, it is needed to assume that despite its dependency on $z$ due to the atmospheric shear, the upstream velocity $U_\infty$ can be considered as a constant when applying the 2D convolution product with the wake centre distribution. For any function $g(y,z)$, this simplification can be written:

$$f_{c,am}(y,z) * *(U_\infty(z) \cdot g(y,z)) = U_\infty(z) \cdot [f_{c,am}(y,z) * *g(y,z)]. \tag{24}$$

An analytical form of the term (I) can then be deduced from Eqs. 15 and 23:

$$(\text{I})_{am}(y,z) = f_{c,am}(y,z) ** [U_\infty(z)(1 - \Delta U_{MF,am}(y,z))]$$

$$= U_\infty(z)\left(1 - \int\int \Delta U_{MF,am}(y-y_c, z-z_c) \cdot f_{c,am}(y_c,z_c)dy_c dz_c\right)$$

$$= U_\infty(z)(1 - \Delta U_{FF,am}) \tag{25}$$

Since it has been shown in the companion paper that term (II) of Eq. 6 is negligible, we do the approximation that $U_{x,FF,am} = (\text{I})_{am}$. The velocity deficit in the FFOR $\Delta U_{FF,am}$ is thus the convolution product of two Gaussian functions. It is known that the convolution product of two normalised Gaussian functions of variance $\sigma_a^2$ and $\sigma_b^2$ is a normalised Gaussian function of variance $\sigma_a^2 + \sigma_b^2$ (Teitelbaum). Equation 25 can be written as the product of two convolution products, leading to:

$$\Delta U_{x,FF,am} = 2C\pi\sigma_y\sigma_z\left[\int \frac{1}{\sqrt{2\pi}\sigma_y}\exp\left(-\frac{(y-y_c)^2}{2\sigma_y^2}\right)\frac{1}{\sqrt{2\pi}\sigma_{fy}}\exp\left(-\frac{y_c^2}{2\sigma_{fy}^2}\right)dy_c\right.$$

$$\left.\cdot\int \frac{1}{\sqrt{2\pi}\sigma_z}\exp\left(-\frac{(z-z_c)^2}{2\sigma_z^2}\right)\frac{1}{\sqrt{2\pi}\sigma_{fz}}\exp\left(-\frac{z_c^2}{2\sigma_{fz}^2}\right)dz_c\right]$$

$$= C\sqrt{\frac{\sigma_y^2}{\sigma_y^2 + \sigma_{fy}^2}\frac{\sigma_z^2}{\sigma_z^2 + \sigma_{fz}^2}}\exp\left(-\frac{y^2}{2\sigma_y^2 + 2\sigma_{fy}^2} - \frac{z^2}{2\sigma_z^2 + 2\sigma_{fz}^2}\right) \tag{26}$$

Even though the reasoning of Braunbehrens and Segalini (2019) is different, it is here shown that their model (Eq. 4) can be found by neglecting term (II) and assuming Eq. 24 as well as Gaussian shapes for the velocity deficit in the MFOR and the wake centre's distribution. This is still a Gaussian form i.e. Eq. 1 with a FFOR wake widths defined as $\sigma_{ty,tz} = \sqrt{\sigma_{y,z}^2 + \sigma_{fy,fz}^2}$, and a maximum velocity deficit of:

$$C_{FF} = C\sqrt{\frac{\sigma_y^2}{\sigma_y^2 + \sigma_{fy}^2}\frac{\sigma_z^2}{\sigma_z^2 + \sigma_{fz}^2}}. \tag{27}$$

To fulfill the conservation of momentum as in Eq. 2, one would need $C_{FF} = 1 - \sqrt{1 - C_T/(8\sigma_{ty}\sigma_{tz}/D^2)}$, which is not the case here. Actually, with this methodology, the conservation of momentum can only be enforced in the MFOR or the FFOR. This is the consequence of neglecting the term (II) in the velocity breakdown, however, the error induced is relatively low since term (II) is negligible. Combining Eqs. 25 and 26 leads to our model for the velocity in the wake of a wind turbine:

$$U_{x,FF,am}(y,z) = U_\infty(z)\left(1 - C\sqrt{\frac{\sigma_y^2}{\sigma_y^2 + \sigma_{fy}^2}\frac{\sigma_z^2}{\sigma_z^2 + \sigma_{fz}^2}}\exp\left(-\frac{y^2}{2\sigma_y^2 + 2\sigma_{fy}^2} - \frac{z^2}{2\sigma_z^2 + 2\sigma_{fz}^2}\right)\right) \tag{28}$$

### 3.3 Model for the turbulence in the FFOR

For the turbulence, a model has been found for terms (III) (Eq. 30) and (IV) (Eq. 33). Even though the contribution of the three cross-terms of Eq. 7 is not always negligible (see companion paper), the two modelled terms are predominant and the result

of the model limited to these two terms can be compared to the turbulence in the FFOR. The total modelled turbulence is here computed as:

$$k_{x,am} = k_{x,m,am} + k_{x,a,am}.$$ 

(29)

### 3.3.1 Meandering term

With the same assumptions as for the term (I), it is possible to derive an analytical formulation for the term (III) of Eq. 7 i.e. the turbulence induced by wake meandering. The assumption of Eq. 24 must again be used to get $U_\infty^2$ out of the convolution product and Eq. 26 is reused to compute the right-hand side of term (III): $\widehat{\overline{U_{MF}}}^2$. On the left-hand side, there is a convolution of the Gaussian function $f_{c,am}$ with $\Delta U_{x,MF,am}^2$, which is also a Gaussian function of widths $\sqrt{0.5}\sigma_y$ and $\sqrt{0.5}\sigma_z$. It is thus possible to use the fact that the convolution of two Gaussian functions is a Gaussian function (Teitelbaum).

$$
\begin{aligned}
(\text{III})_{am} = k_{x,m,am}(y,z) &= \left[ f_{c,am} ** U_{x,MF,am}^2 \right] - U_{x,FF,am}^2 \\
&= U_\infty^2(z) \int\int \left( 1 - \Delta U_{x,MF,am}(y-y_c, z-z_c) \right)^2 f_{c,am}(y_c,z_c) dy_c dz_c - U_\infty^2(z) \left( 1 - \Delta U_{x,FF,am} \right)^2 \\
&= U_\infty^2(z) \int\int \Delta U_{x,MF,am}^2(y-y_c, z-z_c) f_{c,am}(y_c,z_c) dy_c dz_c - U_\infty^2(z) \Delta U_{x,FF,am}^2 \\
&= (CU_\infty(z))^2 \left[ \sqrt{\frac{\sigma_y^2}{\sigma_y^2 + 2\sigma_{fy}^2}} \sqrt{\frac{\sigma_z^2}{\sigma_z^2 + 2\sigma_{fz}^2}} \exp\left( -\frac{y^2}{\sigma_y^2 + 2\sigma_{fy}^2} - \frac{z^2}{\sigma_z^2 + 2\sigma_{fz}^2} \right) \right. \\
&\qquad \left. - \frac{\sigma_y^2}{\sigma_y^2 + \sigma_{fy}^2} \frac{\sigma_z^2}{\sigma_z^2 + \sigma_{fz}^2} \exp\left( -\frac{y^2}{\sigma_y^2 + \sigma_{fy}^2} - \frac{z^2}{\sigma_z^2 + \sigma_{fz}^2} \right) \right]
\end{aligned}
$$

(30)

The shape of term (III) is thus not a double Gaussian, as it may be interpreted in the literature (Stein and Kaltenbach, 2019; Ishihara and Qian, 2018), but rather Gaussian of width $\sqrt{0.5\sigma^2 + \sigma_f^2}$ minus a thinner and less pronounced Gaussian of width $\sqrt{0.5\sigma^2 + 0.5\sigma_f^2}$. It can be verified that this expression is always larger than 0 i.e. the meandering only produces turbulence and does not dissipate it.

### 3.3.2 Rotor-added turbulence term

Combining the chosen models for the wake meandering distribution and the added turbulence in the MFOR (Eqs. 21 and 23 ) in Eq. 20 leads to an analytical form of the axial rotor-added turbulence:

$$(IV)_{am}(y,z) = k_{x,a,am}(y,z) = \max(k_{x,\infty}; k_{x,MF,am} ** f_{c,am})$$

$$= \max\left(k_{x,\infty}; \frac{K_{MF}}{2\pi\sigma_{fy}\sigma_{fz}} \int\int \left[\left(\frac{y_c}{\sigma_y^2}\right)^2 + \left(\frac{z_c}{\sigma_z^2}\right)^2\right] \exp\left(-\frac{y_c^2}{\sigma_y^2} - \frac{z_c^2}{\sigma_z^2}\right) \exp\left(-\frac{(y-y_c)^2}{2\sigma_{fy}^2} - \frac{(z-z_c)^2}{2\sigma_{fz}^2}\right) dy_c dz_c\right)$$

$$= \max\left(k_{x,\infty}; \frac{K_{MF}}{2\pi\sigma_{fy}\sigma_{fz}}\left[\int\left(\frac{y_c}{\sigma_y^2}\right)^2 \exp\left(-\frac{y_c^2}{\sigma_y^2} - \frac{(y-y_c)^2}{2\sigma_{fy}^2}\right) dy_c \int\exp\left(-\frac{z_c^2}{\sigma_z^2} - \frac{(z-z_c)^2}{2\sigma_{fz}^2}\right) dz_c\right.\right.$$

$$\left.\left. + \int\left(\frac{z_c}{\sigma_z^2}\right)^2 \exp\left(-\frac{z_c^2}{\sigma_z^2} - \frac{(z-z_c)^2}{2\sigma_{fz}^2}\right) dz_c \int\exp\left(-\frac{y_c^2}{\sigma_y^2} - \frac{(y-y_c)^2}{2\sigma_{fy}^2}\right) dy_c\right]\right) \quad (31)$$

At this point, the added turbulence in the FFOR is the sum of two terms, that are identical if the coordinates $y$ and $z$ are swapped. It is the product of two convolutions: the first of $f : y \to y^2 \exp(-y^2/\sigma_y^2)$ with a Gaussian function and the second of two Gaussian functions. The first convolution product has been solved with a computer algebra tool (Scherfgen) and the other has already been solved in Eq. 30. It gives:

$$\int\left(\frac{y_c}{\sigma_y^2}\right)^2 \exp\left(-\frac{y_c^2}{\sigma_y^2} - \frac{(y-y_c)^2}{2\sigma_{fy}^2}\right) dy_c \int \exp\left(-\frac{z_c^2}{\sigma_z^2} - \frac{(z-z_c)^2}{2\sigma_{fz}^2}\right) dz_c$$

$$= \frac{\sqrt{2\pi}\sigma_{fy}(\sigma_y^2 y^2 + \sigma_{fy}^4\sigma_y^2 + 2\sigma_{fy}^4)}{\sigma_y(\sigma_y^2 + 2\sigma_{fy}^2)^{5/2}} \exp\left(-\frac{y^2}{\sigma_y^2 + \sigma_{fy}^2}\right) \frac{\sqrt{2\pi}\sigma_{fz}\sigma_z}{\sqrt{\sigma_z^2 + 2\sigma_{fz}^2}} \exp\left(-\frac{z^2}{\sigma_z^2 + 2\sigma_{fz}^2}\right)$$

$$= 2\pi\sigma_{fy}\sigma_{fz}\frac{\sigma_y\sigma_z}{\sqrt{\sigma_y^2 + 2\sigma_{fy}^2}\sqrt{\sigma_z^2 + 2\sigma_{fz}^2}}\frac{(\sigma_y^2 y^2 + \sigma_{fy}^2\sigma_y^2 + 2\sigma_{fy}^4)}{\sigma_y^2(\sigma_y^2 + 2\sigma_{fy}^2)^2} \exp\left(-\frac{y^2}{\sigma_y^2 + \sigma_{fy}^2} - \frac{z^2}{\sigma_z^2 + 2\sigma_{fz}^2}\right) \quad (32)$$

From Eq. 32, it remains to add the same quantity with $y \leftarrow z$ and $z \leftarrow y$, factorise and simplify to deduce the model for $k_{x,a,am}$:

$$k_{x,a,am} = \max\left[k_{x,\infty}; K_{FF}\left(\frac{y^2\sigma_y^2 + \sigma_y^2\sigma_{fy}^2 + 2\sigma_{fy}^4}{\sigma_y^2(\sigma_y^2 + 2\sigma_{fy}^2)^2} + \frac{z^2\sigma_z^2 + \sigma_z^2\sigma_{fz}^2 + 2\sigma_{fz}^4}{\sigma_z^2(\sigma_z^2 + 2\sigma_{fz}^2)^2}\right) \exp\left(-\frac{y^2}{\sigma_y^2 + 2\sigma_{fy}^2} - \frac{z^2}{\sigma_z^2 + 2\sigma_{fz}^2}\right)\right] \quad (33)$$

with:

$$K_{FF} = \frac{K_{MF}}{\sqrt{1 + 2(\sigma_{fy}/\sigma_y)^2}\sqrt{1 + 2(\sigma_{fz}/\sigma_z)^2}}. \quad (34)$$

It can be noted that in the absence of meandering, i.e. for $\sigma_{fy} = \sigma_{fz} = 0$, the model retrieves its MFOR form (Eq. 21). As for the terms (I) and (III), the expression of $k_{x,a,am}$ is based on a Gaussian velocity deficit hypothesis, even in the near wake where the LES wake takes a shape closer to a top-hat function. The velocity gradient that is the source of the rotor-added turbulence is thus lower and more spread in the model compared to the actual values. Another issue of the model is that it poorly takes into account shear, due to the assumptions of Eqs. 20 and 24 . Indeed, the only source of vertical asymmetry in Eq. 33 is $U_\infty^2$, i.e. the velocity shear upstream of the turbine.

## 4 Model's calibration

The model's equations are based on five variables: the wake widths in the MFOR $\sigma_y$ and $\sigma_z$, the modified mixing length $l_m^*$ and the standard deviations of the meandering distribution $\sigma_{fy}$ and $\sigma_{fz}$. Each of these variables needs to be calibrated from the inflow conditions to have a usable model. To do so, the results from the calibration dataset are used. Two versions of the wake meandering calibration: the 'base' calibration, to use if the time series of the upstream velocities are known, and the 'engineering' calibration if they are not.

### 4.1 Wake width in the MFOR

As described in Sec. 3, we assumed that the wake in the MFOR, follows a Gaussian shape function. Moreover, we here assumed that the wake is axisymmetric ($\sigma_y = \sigma_z$) thus reducing the number of parameters in the model from five to four. The width of the wake in the MFOR is deduced from fitting the function of Eq. 35 on the velocity deficit $\Delta U_{MF}$ through a non-linear least squares method.

$$f(y, z, C_0, y_0, z_0, \sigma) = C_0 + C \exp\left(-\frac{(y - y_0)^2}{2\sigma^2} - \frac{(z - z_0)^2}{2\sigma^2}\right) \tag{35}$$

$C$ is fixed as a function of $\sigma$ (Eq. 2 with $\sigma_y = \sigma_z = \sigma$), and the optimisation is run on parameters $\{C_0, y_0, z_0, \sigma\}$ where $y_0, z_0$ are the mean wake centre, $\sigma$ the wake width (the parameter of interest) and $C_0$ is an offset to help the algorithm.

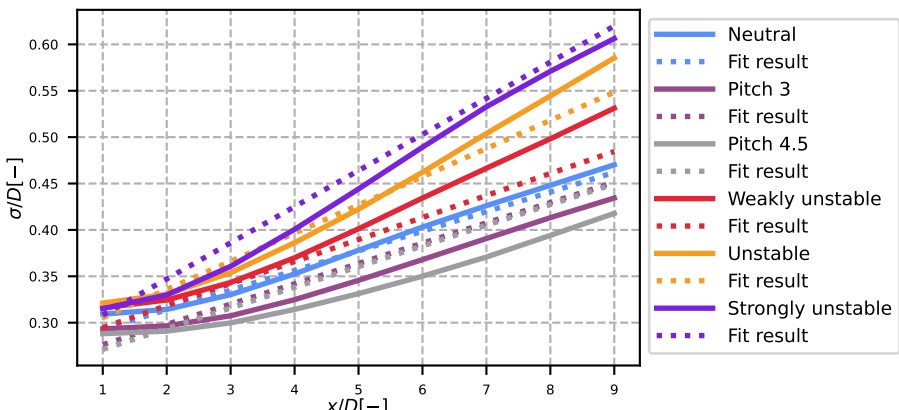

**Figure 5.** Wake width in the MFOR for the different cases of the calibration dataset. Solid lines: results from the LES simulation (Eq. 35; dotted lines: proposed calibration.

The resulting wake widths in the MFOR as a function of the downstream distance are plotted in solid lines in Fig. 5 for the six cases of the calibration dataset. Excepted in the near wake, the wake width evolves linearly with the distance to the turbine. Moreover, the greater the unstability (and thus the level of turbulence, cf Fig. 2), the greater the slope of this linear relation. Finally, the simulations with degraded thrust seem to have the same slope as the neutral case, but with a different origin.

For all these reasons, the chosen function for the calibration is the following:

$$\sigma/D = (aI + b)\frac{x}{D} + c\sqrt{\beta} \tag{36}$$

where $a$, $b$ and $c$ are parameters to fit, $I$ is the total turbulence intensity (Eq. 9) and $\beta = 0.5\left(1 + \sqrt{1 - C_t}\right)/\sqrt{1 - C_t}$ (Bastankhah and Porté-Agel, 2014). A least square fit method on the six different curves allowed to compute the best values

of $a$, $b$ and $c$ (see Table. 3). Note that this fit is in the end very similar to what can be found in the literature (e.g. Fuertes et al. (2018)), except that the slope (parameter $a$) is smaller because the models of the literature implicitly assume that the meandering is included in the wake expansion.

| Parameter | a [-] | b [-] | c [-] |
|-----------|-------|-------|-------|
| Value | 0.276 | -0.00329 | 0.231 |

Table 3. Parameters for the wake width in the MFOR.

## 4.2 Modified mixing length

The modified mixing length $l_m^*$ in Eq. 33 directly drives the amount of turbulence added by the turbine. In Sect. 3, it was

385 shown that this variable in the upper part of the wake is independent of the simulation case when normalised with the upstream modified mixing length. Therefore, the evolution of $l_m^*/l_{m,\infty}^*$ has been plotted in Fig. 6 and it shows an approximately linear behaviour with the downstream distance.

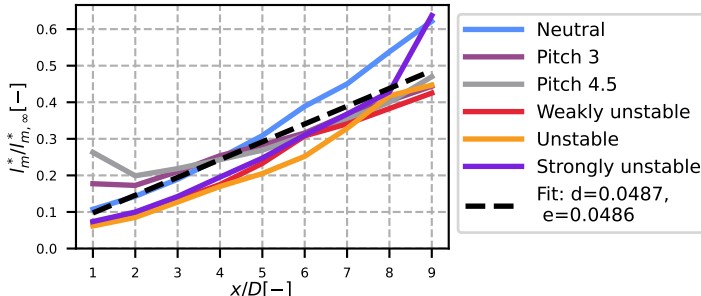

Figure 6. Normalised modified mixing length for the different cases of the calibration dataset.

In first approach, it is thus decided to fit the mixing length with a linear function of $x/D$:

$$l_m^* = l_{m,\infty}^*\left(d\frac{x}{D} + e\right) \tag{37}$$

where $l_{m,\infty}^*$ is deduced from Eqs. 13 and 14, in which $u_*$ and $L_{MO}$ can be found from a fit of the inflow velocity profile. A least square fit method on the different curves from Fig. 6 is used to fit Eq. 37. The resulting parameters $d$ and $e$ can be found

in Table. 4 and the corresponding fitted function is plotted in dashed black line in Fig. 6. The results are quite satisfying even though all the curves are not perfectly superimposed.

| Parameter | d [-] | e [-] |
|-----------|--------|--------|
| Value | 0.0487 | 0.0486 |

**Table 4.** Parameters for the mixing length.

## 4.3 Wake meandering

The widths of the wake centre's distribution $\sigma_{fy}$ and $\sigma_{fz}$ are computed as the standard deviations of the wake centre's coordinate $y_c(x,t)$ and $z_c(x,t)$:

$$\sigma_{fy}(x) = \sqrt{\overline{y_c(x,t)'^2}} \; ; \quad \sigma_{fz}(x) = \sqrt{\overline{z_c(x,t)'^2}} \tag{38}$$

The resulting amount of meandering in the horizontal (top figure) and vertical direction (bottom figure) for the six cases of the calibration dataset can be found in Fig. 7. The LES results are plotted in solid lines. Overall, the more unstable the case, the

400 more meandering is found. However, the meandering does not solely depend on the lateral turbulence intensity. In particular, the weakly unstable case has greater vertical meandering than the unstable case, despite having a lower $I_z$ value (see Table. 1). It is also worth noting that the reduction of the thrust coefficient have little to no effect on the meandering (all the neutral cases are equivalent).

To model the amount of meandering, Braunbehrens and Segalini (2019) propose the following formula:

$$\sigma_{fy,fz}(x)^2 = 2k_{y,z} \int\limits_0^{x/U_c} (\frac{x}{U_c} - \zeta)\mathcal{A}_{v,w}(\zeta)d\zeta \tag{39}$$

where $U_c$ is an advection velocity and $\mathcal{A}$ is the autocorrelation function of the velocity (respectively the lateral and vertical one). For each case, the results of Eq 39 are plotted in dashed line in Fig. 7 for $U_c = 0.8U_\infty$. This model for the amount of meandering works fairly well, with the right order of magnitude in each case, and it predicts the different behaviour of the vertical and lateral directions for the unstable and weakly unstable cases. However, such calibration for $\sigma_{fy}$ and $\sigma_{fz}$ is not

appropriate for analytical wake modelling because it requires time series of wind velocities at hub height whereas usually, only the mean values are available.

Therefore, we propose hereby (dotted lines in Fig. 7) an engineering-oriented solution to approximate the amount of meandering without access to the unsteady time series of velocities upstream of the turbine. In the first attempts to model the meandering (Ainslie, 1988), it was proposed that the wake meandering should be a linear function of the inflow wind direc-

415 tion's variance. However, more recent work (Doubrawa et al., 2018) showed that the amount of meandering decreases with

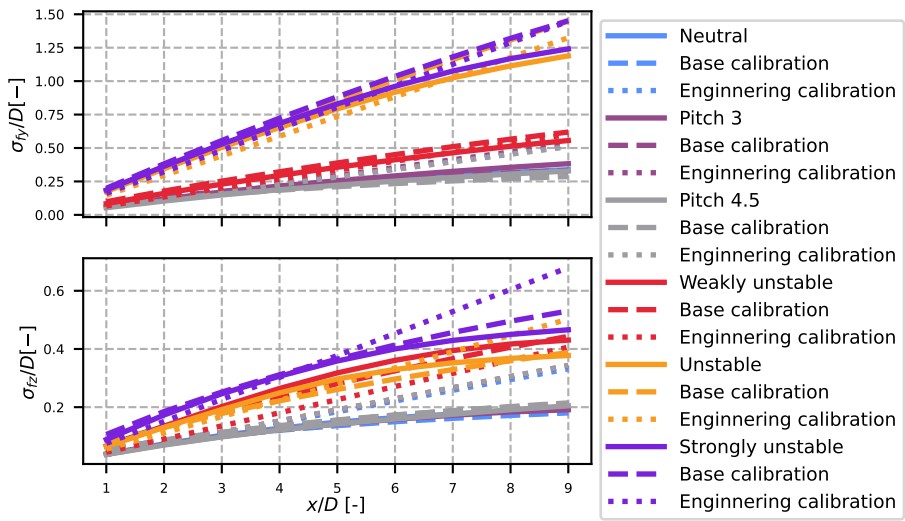

**Figure 7.** Normalised standard deviation of the wake centre from the LES (solid lines), results from the base calibration (dashed line) and from the engineering calibration (dotted line).

the rotor size. Indeed, following the theory of the DWM model, only the eddies larger than the size of the rotor are energetic enough to induce wake meandering. Thus the idea is to calibrate the amount of wake meandering only with eddies larger than this size:

$$\sigma_{fy} = \frac{\sqrt{k_y^D}}{U_\infty} \frac{x}{D} \tag{40}$$

and similarly for $\sigma_{fz}$. In Eq. 40, $k_y^D$ is the lateral turbulence with size larger than the diameter of the turbine, i.e. the variance of the wind velocity averaged over a circle of two rotor diameters and centred at the hub. Note that the time variance is performed after the spatial averaging.

The issue is that $k_y^D$ and $k_z^D$ are not known *a priori*, and since the stability of the ABL modifies the low-frequency range of the turbulence spectrum, it is expected that the share of the turbulence with larger size than the rotor to the total turbulence

is dependent on the atmospheric stability. This can be observed in Fig. 8 where the ratio between the turbulence larger than a disk of diameter $d_{disk}$, $k_{y,z}^{d_{disk}}$ to the total turbulence is computed for $k_y$ and $k_z$ for each case.

Figure 8 highlights two distinct behaviours, depending on the stability conditions: the unstable cases (orange and purple curves) decrease much slower than the near neutral cases (red, grey, brown and blue), and this phenomenon is particularly marked for the lateral turbulence. It shows that the unstable cases have (in proportion) more low-frequency (or large-size

eddies) than the near-neutral cases.

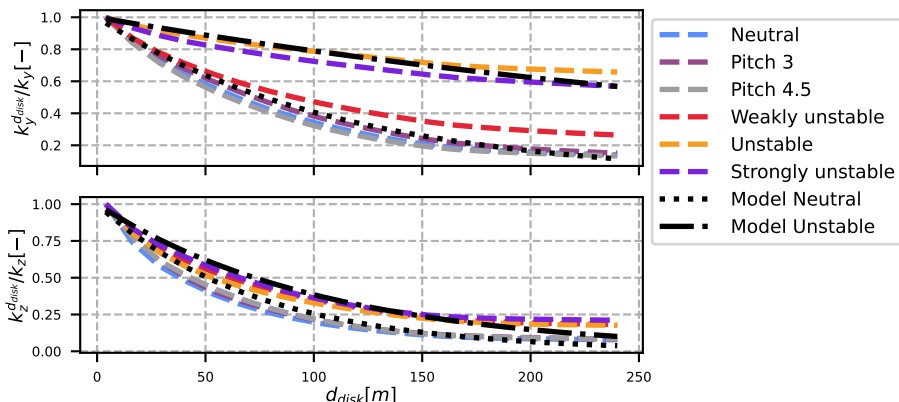

**Figure 8.** Ratio of turbulence averaged over a disk to the total turbulence, for different disk sizes.

Even though a fully physical approach would require a measure of the stability and an in-depth study of the turbulence spectrum in function of the ABL conditions, the objective here is to propose an analytical model easy to implement and use. It is thus proposed to model the ratio $k_y^{d_{disk}}/k_y$ and $k_z^{d_{disk}}/k_z$ with an analytical function:

$$k_y^d/k_y = \exp\left(-d_{disk}/\Gamma_y\right)$$

$$k_z^d/k_z = \exp\left(-d_{disk}/\Gamma_z\right) \tag{41}$$

A least square fit has been used to determine the value of the parameter $\Gamma$. Two different fits were used in order to have one result for unstable cases and one for near-neutral cases. The results are given in Table and the Eq. 41 is plotted in Figure. 8 in black dotted and black dash-dotted lines for the neutral and unstable values, respectively.

| Case | $\Gamma_y[m]$ | $\Gamma_z[m]$ |
|------|---------------|---------------|
| Neutral | 56 | 37 |
| Unstable | 212 | 52 |

**Table 5.** Parameters for the large scale turbulence function.

$\Gamma$ can be interpreted as a measure of the large-scale eddies of the atmosphere, even though it is not defined as the integral

length scale. The combination of Eqs. 40 and 41 with values from Table. 5 is plotted in dotted lines in Fig. 7. Even though the model cannot predict the non-linear behaviour in the far wake, the results remain quite good. Only the weakly unstable case gives poor results, allegedly because it is at the edge between the near neutral and unstable case, and would necessitate a value of $\Gamma$ of his own.

## 5 Results

In this section, we analyse the results of the new model described in the precedent sections. For the streamwise velocity, the model is described with Eq. 28 and for streamwise turbulence with the sum of Eqs. 30 and 33. This validation is done with the two validation cases (see Table. 1), i.e. with the unstable and neutral SWiFT simulations. Three versions of the calibration of $\sigma$, $\sigma_{fy}$, $\sigma_{fz}$ and $l_m^*$ are shown:

- The 'base' calibration is defined with Eqs. 36, 38 and 39 and values for $a$, $b$, $c$, $d$ and $e$ from Tables. 3 and 4. This calibration makes more sense physically but requires the time series of the inflow velocities to determine the autocorrelation necessary to compute $\sigma_f$ from Eq. 39. It is plotted in blue dashed lines in Figs. 9 to 12.

- The 'engineering' calibration uses the same equations except for the wake meandering, where Eqs. 40 and 41 are used instead of Eq. 38 and parameters $\Gamma$ are taken from Table. 5. It is plotted in red dotted lines in Figs. 9 to 12.

- Finally, we also proposed the 'best' version of the model. Knowing that the calibration produces errors, it seemed interesting to see what would be the results of the 'best calibration possible', i.e. with parameters $\sigma$, $\sigma_{fy}$, $\sigma_{fz}$ and $l_m^*$ directly taken from the LES simulation of the SWiFT simulation (and not from the calibration deduced from the Sect. 4). Obviously, this version of the model cannot be used, but it is helpful to determine if the discrepancies between our model and the LES come from the calibration or the construction of the model itself. It is plotted in orange dash-dotted lines in Figs. 9 to 12.

Additionally, the reader will find in the following figures the results directly from Meso-NH (in black solid line) and the result from a widely used model of the I&Q2018 model (in purple dash-dot-dotted line), one of the few in the literature that predict both profiles of mean streamwise velocity and streamwise turbulence.

### 5.1 Velocity field

The results for the streamwise velocity field in the FFOR can be found in Figs. 9 and 10 for the neutral and unstable cases, respectively. The horizontal (top) and vertical (bottom) profiles of velocity are plotted for the reference LES, results from the literature, and the three versions of the aforementioned model's calibration. The three columns are three different positions downstream of the wind turbine: $x/D = 2$, $x/D = 5$ and $x/D = 8$.

Our model (with any calibration) behaves very similarly to the I&Q2018 model in the neutral case (Fig. 9), and both are accurate compared to the LES data in black. The only discrepancy is in the near wake, where both models assume a Gaussian shape whereas a super-Gaussian shape (Blondel and Cathelain, 2020) would be more appropriate. These overall good results confirm that the hypotheses made in Sect. 3 for the velocity in the MFOR and the wake centre distribution are good and that meandering has been correctly computed.

In the unstable case (Fig. 10), the literature model underestimates the wake dissipation, whereas the proposed model is more accurate. This is because the I&Q2018 model only uses $C_T$ and $I_x$ as parameters. As shown in Table. 1, these values are very

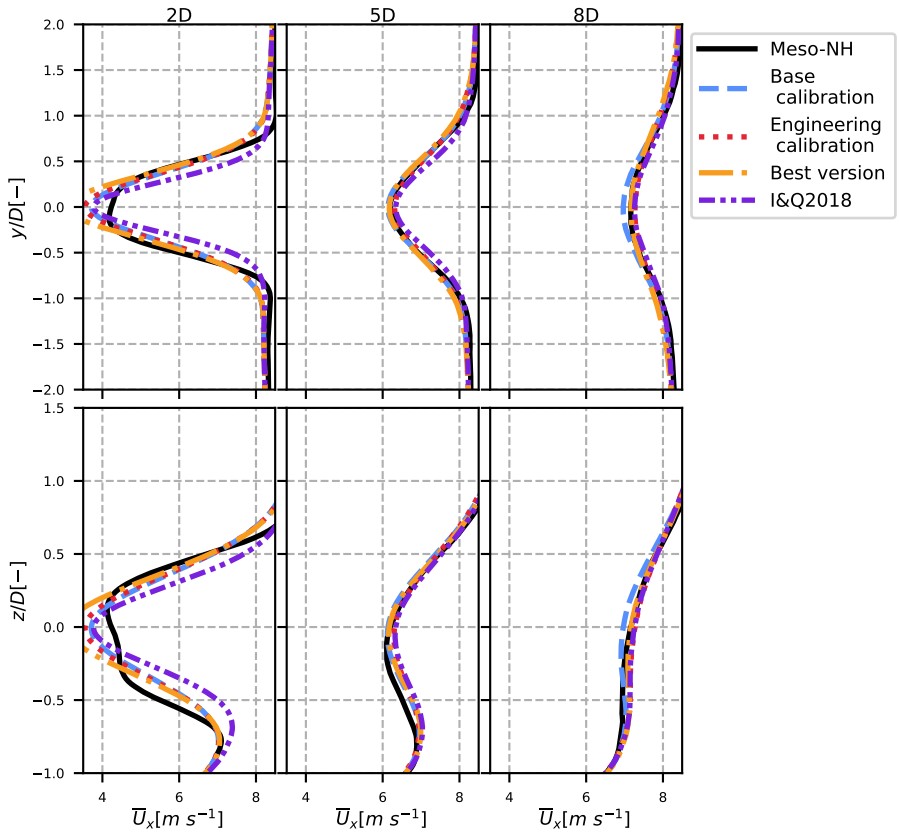

**Figure 9.** Results of the analytical velocity model for the different calibrations (blue dashed, red dotted and orange dash-dotted lines) in the neutral case, compared to Meso-NH (in black solid line) and the I&Q2018 model (red dotted line). Lateral (top) and vertical (bottom) profiles are plotted for different positions downstream.

similar in the neutral and unstable cases of the validation case, and thus the I&Q2018 results are very similar between the neutral and unstable cases. It cannot predict the increase of meandering under unstable ABL due to higher values of large-scale turbulence in the lateral and vertical directions.

The proposed model is better on that matter, showing a larger wake expansion due to the higher predicted meandering compared to the neutral case. It shows that the determination of the velocity deficit in non-neutral cases necessitates more than only the total streamwise turbulence. In this case, one can note a discrepancy between the 'best version' and the two calibrations of our model. It is due to an overestimation of the wake width in the MFOR for the unstable case (not shown here). Indeed, the neutral and unstable cases have similar wake widths in the MFOR while having different total turbulence intensities $I$ (Table. 1), and therefore Eq. 36 gives accurate results for the neutral case but overestimates the MFOR wake width in the unstable case. As a result, there is a compensation of error, where the calibration underestimates the velocity deficit whereas

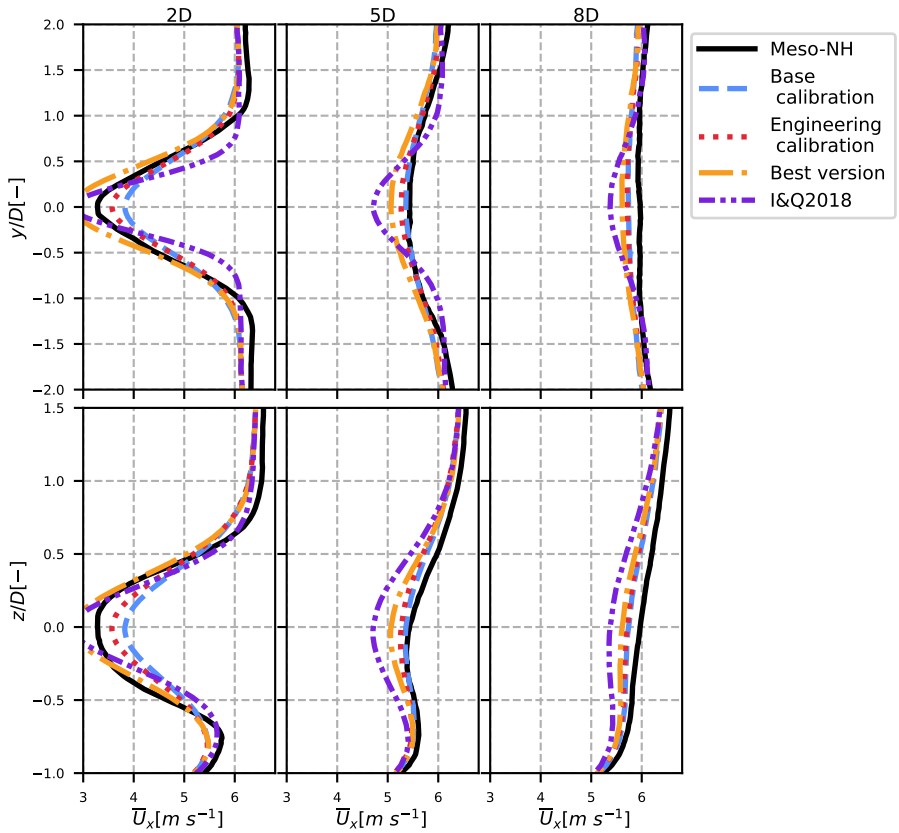

**Figure 10.** Results of the analytical velocity model for the different calibrations (blue dashed, red dotted and orange dash-dotted lines) in the unstable case, compared to Meso-NH (in black solid line) and the I&Q2018 model (red dotted line). Lateral (top) and vertical (bottom) profiles are plotted for different positions downstream.

the 'best' version is supposed to slightly overestimate it, resulting in a very good match. Nevertheless, even without this error compensation, the 'best' version still outperforms the literature model.

## 5.2 Turbulence field

With the same plotting convention as in Figs. 9 and 10, the profiles of turbulence in the horizontal and vertical directions are plotted in Figs. 11 and 12 for the neutral and unstable cases, respectively.

In the neutral case (Fig. 11), the I&Q2018 model is performing remarkably well. It correctly predicts the location of the double peak in the horizontal direction and of the top tip peak in the vertical direction. The proposed model shows less good results: despite the order of magnitude being accurate, the shape of the function is not, and the top-tip maximum is not correctly positioned. Since the calibrations do not significantly differ from the 'best version' of the model, this is attributed to modelling

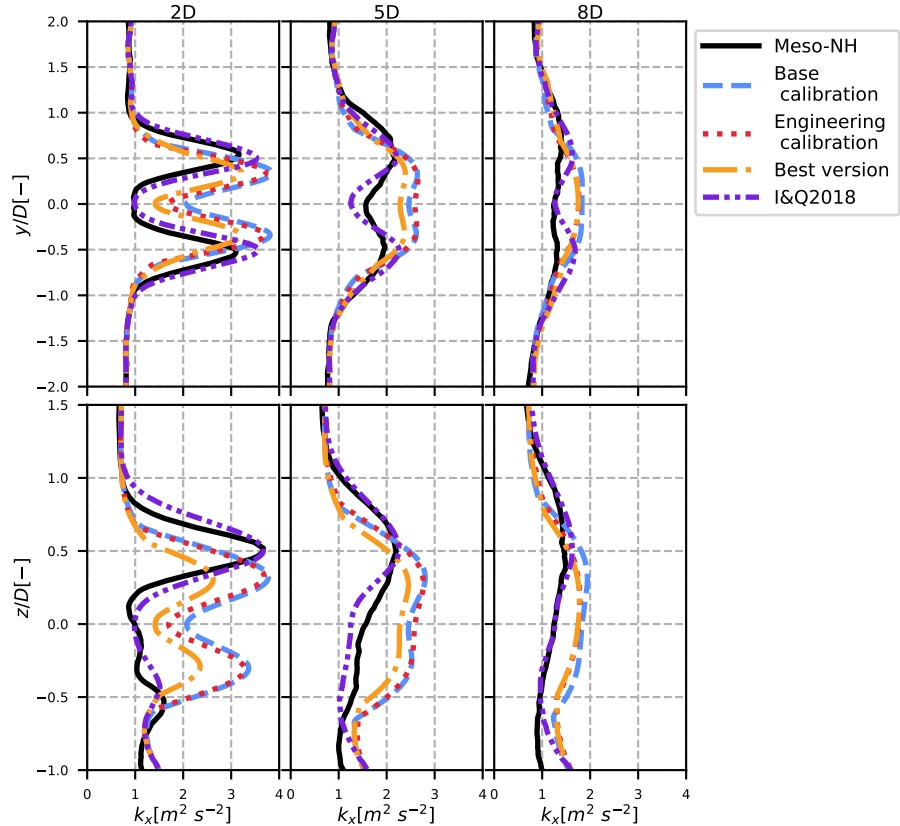

**Figure 11.** Results of the analytical streamwise turbulence model for the different calibrations (blue dashed, red dotted and orange dash-dotted lines) in the neutral case, compared to Meso-NH (in black solid line) and the I&Q2018 model (red dotted line). Lateral (top) and vertical (bottom) profiles are plotted for different positions downstream.

errors, and not to the calibration. The authors suggest that this deviation originates from the omission of shear in the modeling of rotor-added turbulence, as described in Equation 33.

The unstable case shows the main shortcomings of the I&Q2018 model and the added value of our model. As shown previously, the I&Q2018 model gives similar results between the unstable and neutral SWiFT cases because they have similar inflow $I_x$ and $C_T$ values. However, the Meso-NH simulations show significant differences, in particular the fact that around $x = 5D$, the turbulence profile is unimodal in the unstable case and bimodal in the neutral case. This difference cannot be predicted by the I&Q2018 model as it assumes always a bimodal shape with a maximum at the top tip. However, this change of shape can be predicted by our model since both Eqs. 30 and 33 are bimodal when $\sigma >> \sigma_f$ and unimodal when $\sigma << \sigma_f$.

Except for the upstream turbulence profiles, the inflow conditions used in the I&Q2018 are very similar between the neutral and unstable cases. Consequently, the purple profiles are alike in Figs. 11 and 12 whereas the stronger meandering in the unstable case leads to a Gaussian-like turbulence profile, even in the vertical direction. The maximum turbulence is thus no

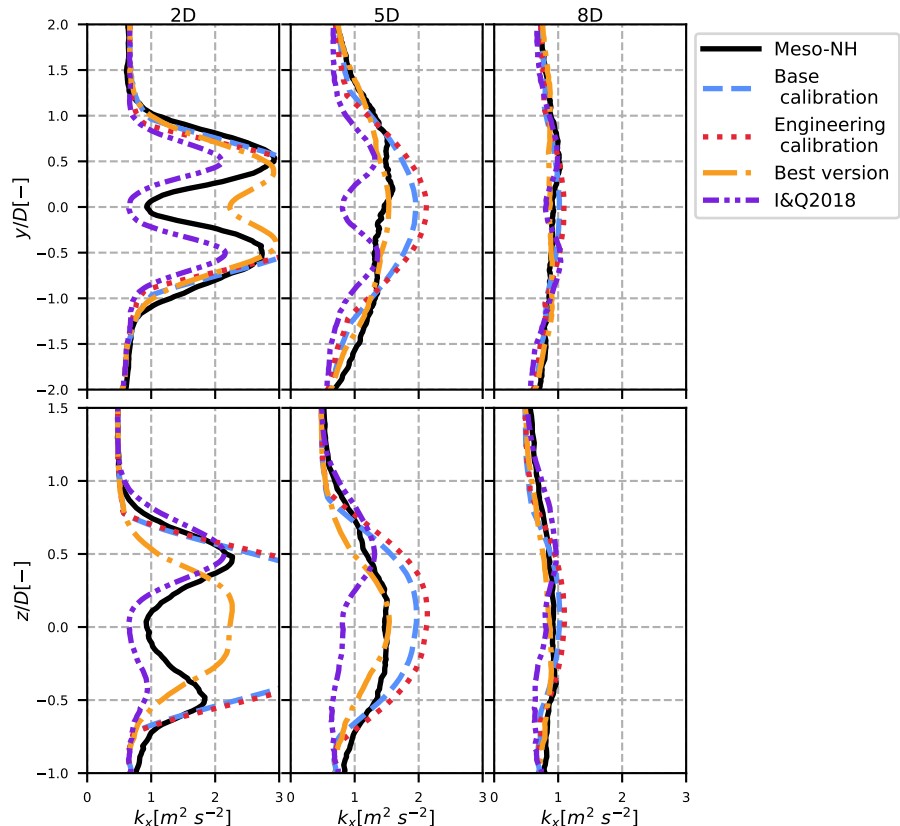

**Figure 12.** Results of the analytical streamwise turbulence model for the different calibrations (blue dashed, red dotted and orange dash-dotted lines) in the unstable case, compared to Meso-NH (in black solid line) and the I&Q2018 model (red dotted line). Lateral (top) and vertical (bottom) profiles are plotted for different positions downstream.

longer located at the top tip but rather at hub height. This property is well-predicted by our model but not by the I&Q2018 model, which does not take meandering into account, and predicts quasi-identical behaviours between the neutral and unstable cases. As shown in Figs. 5 and 7, the amount of meandering starts lower but grows faster than the wake width in the MFOR, in particular in unstable conditions. Hence, one can expect that a bimodal shape in the near wake and an unimodal shape in the far wake, as seen in Figs. 11 and 12.

However, the calibration of our model leads to an overestimation of the streamwise turbulence, in particular in the near wake. Since there are not many differences between the basic and engineering calibrations, it is not attributed to the meandering calibration (these two calibrations only differ by the meandering modelling), but rather to the overestimated $\sigma$ in the MFOR, as well as an overestimated $l_m^*$. When computed directly from the simulation, the values of $l_m^*$ are very similar between the neutral and unstable cases, whereas the values of $l_{m,\infty}^*$ are much greater in the unstable case (see Fig. 3). Therefore, the value of $l_m^*$ is

overestimated by the model, leading to an overestimation of the rotor-added turbulence, and thus to the total turbulence.

The 'best version' of the model gives interesting results, showing that if a better calibration was achieved, in particular for the modified mixing length, the results of the model would be better. This question will be further detailed in the next section.

## 6 Discussion

The previous section showed the results of the model developed in this paper. It is quite good for the streamwise velocity field but can be improved for the turbulence, where the fully empirical model of Ishihara and Qian (2018) shows overall better results in neutral cases but has shortcomings in unstable cases. However, since it is physically-based, we know the assumptions of the present model and thus have clear possibilities for improvements. The main ones known by the authors are listed below. Moreover, this work shows that the modification of the velocity and turbulence fields when the ABL stability is modified (and not the $I_x$ or $C_T$) can be predicted. This is a crucial point, as future applications of analytical models such as digital twins will require an estimation of the wake velocity and turbulence over small time-lapses and not a yearly average like AEP calculations.

The authors want to emphasise that the presented work is a first step toward a fully physically-based model for turbulence profiles that depend on atmospheric stability. In the companion paper, it was shown that the turbulence in the wake of a wind turbine is the sum of several terms, and here we presented a methodology to model analytically the most important of these terms. Even though a fully usable calibration is proposed for anyone who would like to test the model, the main purpose of this work is to demonstrate how the rotor-added turbulence and meandering turbulence can be modelled from simple functions.

### 6.1 Calibration improvement

In Figs. 9 to 12, there are discrepancies between the 'best version' of the model and our proposed calibrations. This is particularly true for turbulence, and it is attributed to the calibration of $l_m^*$. Contrarily to $\sigma$ and $\sigma_f$ which can be computed on a wake no matter what, our computation of the modified mixing length $l_m^*$ makes sense only if it is assumed that the rotor-added turbulence only comes from the wake shear. Additionally, the vertical velocity gradient of the ABL $\partial U_\infty / \partial z$ is voluntarily omitted in Eq. 20.

On one hand, all of these assumptions make the measure of $l_m^*$ a hardly reliable variable. On the other hand, our model is strongly dependent on this parameter. Indeed, the rotor-added turbulence is proportional to the square of $l_m^*$. Therefore, a small over- or underestimation of $l_m^*$ is likely to happen and it leads to large differences. In Fig. 13 is shown the effect of multiplying parameter $d$ of the mixing length (Table. 4) by a factor $0.8$ (red dotted line), $1.2$ (orange dash-dotted line) and $1.5$ (purple dash-dot-dotted line) for the 'basic' calibration, in the neutral case. It results in large differences from one result to another, showing that even small differences in $l_m^*$ can drastically change the conclusions.

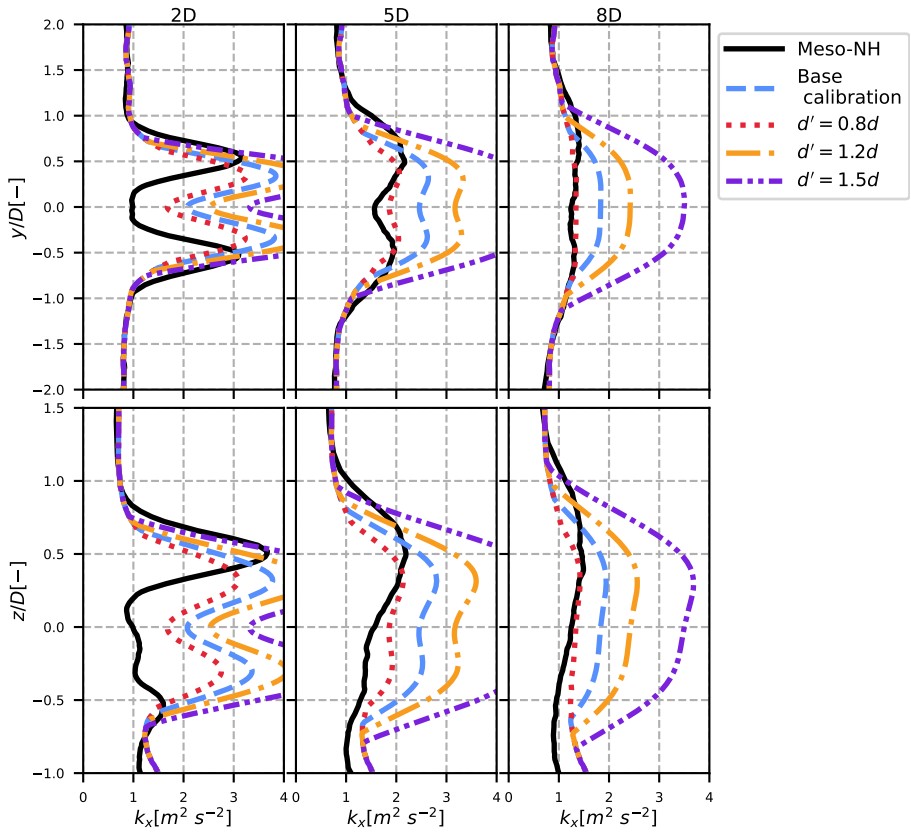

**Figure 13.** Results of the axial turbulence analytical velocity model in the neutral case, for different values of parameter $d$ in the calibration of $l_m^*$.

## 6.2 Modelisation improvements

Besides a better calibration, the model could benefit from conceptual improvements. Indeed, the 'best version' of the model (orange curve in Figs. 11 and 12) does not match the LES results. In other words, even with a 'perfect' calibration, the model still misses some features of the turbulence in the wake.

At several points of the reasoning, the atmospheric shear, i.e. the dependence of $U_\infty$ with $z$ is neglected (Eqs. 24 and 20). The first improvement that comes to mind is to model the interaction between atmospheric and wake shear. By doing so, it would be possible to have the reduction of shear near the ground and an increase of shear at the top tip, leading to a smaller value of turbulence at the bottom tip compared to the top tip, as observed in the LES datasets and modelled in the I&Q2018 model. In the model under its current form, the shear is only accounted for through $U_\infty^2$ in factor of $k_{m,am}$ and $k_{a,am}$. This small contribution is compensated by the upstream turbulence $k_\infty$ that is larger at the bottom than at the top, leading to almost symmetric vertical profiles for the model whereas the LES profiles.

A second improvement that could be done concerns the near wake. As mentioned in Sect. 5, instead of using a simple Gaussian function, a super-Gaussian function would be more accurate. This generic function takes a top-hat form in the near wake and progressively transitions to a Gaussian function as it travels downstream. It was shown in Blondel and Cathelain (2020) that it gives more accurate results in the near wake. Such a function would not only improve the velocity model but also the meandering and rotor-added turbulence terms, which are built upon the velocity model. The latter in particular is a function

of the spatial derivative of $\Delta U$: using the Gaussian function instead of the super-Gaussian function as done in this work thus leading to an underestimation of the shear at the edge of the turbine.

    For both of these improvements, some solutions were tried: not neglecting the $\partial U_\infty / \partial z$ in the derivation of the rotor added turbulence and using a super-Gaussian function instead of a Gaussian for the velocity in the MFOR. In both cases, no analytical solution for the models was reached. If such a fully-analytical resolution is indeed impossible, an approximated form (for

instance based on LES results) could be proposed in the future.

    Finally, modelling the additional terms of Eq. 7, in particular the covariance term (V) could further improve the model. It was shown in the companion paper that this term can represent about 10% of the total turbulence in the wake and redistributes the turbulence vertically. Given the order of magnitude, this is of lesser importance than the points aforementioned, but would also improve the results, or at least the physical accuracy of the model.

## 7   Conclusions

This work is the second part of a two-step study that aims at modelling the turbulence in the wake of a wind turbine based on the meandering phenomenon. In the companion paper, the velocity and turbulence in the FFOR were broken down into different terms, some of which were shown to be negligible. In the present work, an analytical model is proposed for the dominating terms of the velocity and turbulence breakdowns, i.e. the meandering turbulence and the rotor-added turbulence.

The originality of this work is that it allows modelling independently the effects of meandering (and thus of the ABL stability) and the wake expansion and that it gives the whole turbulence profile rather than only the maximum value. For the velocity, it writes:

$$U_{x,am}(y,z) = U_\infty(z) \left( 1 - C \sqrt{\frac{\sigma_y^2}{\sigma_y^2 + \sigma_{fy}^2} \frac{\sigma_z^2}{\sigma_z^2 + \sigma_{fz}^2}} \exp\left( -\frac{y^2}{2\sigma_y^2 + 2\sigma_{fy}^2} - \frac{z^2}{2\sigma_z^2 + 2\sigma_{fz}^2} \right) \right) \tag{42}$$

and for the turbulence:

$$580 \quad k_{x,am} = \max \Bigg[ k_{x,\infty},$$

$$\frac{(CU_\infty(z)l_m^*(x))^2}{\sqrt{1+2(\sigma_{fy}/\sigma_y)^2}\sqrt{1+2(\sigma_{fz}/\sigma_z)^2}} \left( \frac{y^2\sigma_y^2 + \sigma_y^2\sigma_{fy}^2 + 2\sigma_{fy}^4}{\sigma_y^2(\sigma_y^2 + 2\sigma_{fy}^2)^2} + \frac{z^2\sigma_z^2 + \sigma_z^2\sigma_{fz}^2 + 2\sigma_{fz}^4}{\sigma_z^2(\sigma_z^2 + 2\sigma_{fz}^2)^2} \right) \exp\left( -\frac{y^2}{\sigma_y^2 + 2\sigma_{fy}^2} - \frac{z^2}{\sigma_z^2 + 2\sigma_{fz}^2} \right) \Bigg] +$$

$$(CU_\infty(z))^2 \left[ \sqrt{\frac{\sigma_y^2}{\sigma_y^2 + 2\sigma_{fy}^2}} \sqrt{\frac{\sigma_z^2}{\sigma_z^2 + 2\sigma_{fz}^2}} \exp\left( -\frac{y^2}{\sigma_y^2 + 2\sigma_{fy}^2} - \frac{z^2}{\sigma_z^2 + 2\sigma_{fz}^2} \right) - \frac{\sigma_y^2}{\sigma_y^2 + \sigma_{fy}^2} \frac{\sigma_z^2}{\sigma_z^2 + \sigma_{fz}^2} \exp\left( -\frac{y^2}{\sigma_y^2 + \sigma_{fy}^2} - \frac{z^2}{\sigma_z^2 + \sigma_{fz}^2} \right) \right) \Bigg]$$

$$\tag{43}$$

where $C = 1 - \sqrt{1 - C_T/(8\sigma_y\sigma_z/D^2)}$, $C_T$ is the thrust coefficient, $D$ is the turbine diameter, $k_{x,\infty}$ and $U_\infty$ are the variance and mean values of the upstream axial velocity. The model's parameters are the wake widths $\sigma_y, \sigma_z$, the amount of meandering $\sigma_{fy}, \sigma_{fz}$ and the modified mixing length $l_m^*$. Two calibrations of these parameters are proposed in Table. 6: the first one ('base' calibration) can be used if time series of the wind velocity are available and the second one ('engineering' calibration) if they are not. In this table, $\mathcal{A}_\phi$ is the autocorrelation of $\phi$, $U_c = 0.8U_\infty$ and $l_{m,\infty}^*$ is found by fitting the inflow velocity profile (Eq. 14). The expressions of velocity and added turbulence in the MFOR used to build Eqs. 42 and 43 can also be used as inputs to the DWM: combined with a synthetic turbulence generation, the unsteady effects of meandering can be modelled.

The model has been tested on two LESs simulations of a single wind turbine wake under a neutral and unstable atmosphere. For the velocity, the results are satisfactory, either in the vertical or lateral direction. The model performs better than the model from Ishihara and Qian (2018) in the unstable case as it predicts correctly the increased dissipation due to the increase of meandering. For the turbulence profiles, however, the results are not as good. Since the atmospheric shear was neglected in several steps of the model, the maximum turbulence at the top tip in the neutral case could not be predicted. In the unstable case, the modified mixing length $l_m^*$ was overestimated and since the model is very sensitive to this parameter, it resulted in too large values of added turbulence. However, the model of Ishihara and Qian (2018) does not predict correctly the turbulence in the unstable case either. In particular, it still predicts a bimodal shape with a maximum at the top tip in all the wake, whereas the proposed model successfully transitions from a bimodal to an unimodal shape, according to the LES results.

| Calibration | $\sigma_y/D = \sigma_z/D = \sigma/D$ | | $l_m^*$ | | $\sigma_{fy}/D$ | $\sigma_{fz}/D$ |
|---|---|---|---|---|---|---|
| Base | $(aI+b)\dfrac{x}{D} + c\sqrt{\beta}$ | | $l_{m,\infty}^*\left(d\dfrac{x}{D}+e\right)$ | | $\sqrt{2k_y \int_0^{x/U_c}(\frac{x}{U_c}-\zeta)\mathcal{A}_v(\zeta)d\zeta}$ | $\sqrt{2k_z \int_0^{x/U_c}(\frac{x}{U_c}-\zeta)\mathcal{A}_w(\zeta)d\zeta}$ |
| Engineering | $(aI+b)\dfrac{x}{D} + c\sqrt{\beta}$ | | $l_{m,\infty}^*\left(d\dfrac{x}{D}+e\right)$ | | $\dfrac{\sqrt{k_y\exp(-D/\Gamma_y)}}{U_\infty}\dfrac{x}{D}$ | $\dfrac{\sqrt{k_y\exp(-D/\Gamma_z)}}{U_\infty}\dfrac{x}{D}$ |
| | a | b | c | d | e | $\Gamma_y$ | $\Gamma_z$ |
| Value | 0.276 | -0.00329 | 0.231 | 0.0487 | 0.0486 | Neutral: 56m <br> Unstable: 212m | Neutral: 37m <br> Unstable: 52m |

**Table 6.** Calibration's parameters of the model

This is the first step toward a fully analytical, physically-based model for turbulence and velocity profiles in the wake of a wind turbine that takes into account atmospheric stability. For future works, the treatment of shear must be improved to

model more realistically vertical turbulence profiles. The MFOR velocity deficit function could be replaced by a more accurate function in the near wake to improve the model's results in this region. It would also be interesting to derive an analytical model for the other terms of the turbulence breakdown.

Finally, this model can currently only be used for one turbine, as it predicts only the streamwise velocity and turbulence, but necessitates the upstream lateral and vertical turbulence. For the model to be usable for multi-turbines, an expression for every term of the Reynolds-stress tensor (or at least the diagonal terms to get the total TKE) would be needed, which implies a model for the lateral and vertical velocities $U_y$ and $U_z$. This also implies more advanced studies of wake meandering from a turbine working in waked conditions, as most of the wake meandering studies are performed in freestream conditions.

*Code and data availability.* The code Meso-NH is open-source and can be downloaded on the dedicated website. The authors can provide the source code of the modified version 5-4-3 that was used in this work. The data used for the plot presented here and in part 1 are available under this online deposit: 10.5281/zenodo.6562720. The data for the calibration can be found under the following DOI: https://dx.doi.org/10.25326/568.. The model equations have been written in python under the following online deposit 10.5281/zenodo.10245174.

*Author contributions.* EJ wrote the analytical model with FB. All the authors worked on the interpretation of the results. The manuscript has been written by EJ with the feedbacks of FB and VM.

*Competing interests.* The authors declare that they have no competing interests.

*Acknowledgements.* The authors would like to thank the different stakeholders of the MOMENTA project (Jézéquel, 2023) for allowing to use the LES data for the calibration of the present model.

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
