# Peer review of "Breakdown of the velocity and turbulence in the wake of a wind turbine - Part 2: Analytical modeling."

_Wind Energy Science, 2022_

## Referee Comment (RC2)

**REVIEW OF WES-2022-47**

*Breakdown of the velocity and turbulence in the wake of a wind turbine - Part 2: Analytical modeling.*

*authors:*
Erwan Jézéquel
Frederic Blondel
Valery Masson

**Summary:**

The manuscript entitled "Breakdown of the velocity and turbulence in the wake of a wind turbine - Part 2: Analytical modeling" endeavors to describe the turbulent velocity field in a wind turbine wake by accounting for energy in both the meandering and fixed frames of reference. The mathematical development is thorough and detailed, even if the presentation is difficult to follow at times. The model development takes cues from some well known approaches in fundamental turbulence (e.g., the Boussinesq Hypothesis) and is almost entirely analytical, leaving very few constants to be tuned empirically. For the most part, the manuscript is well written and clear, although there are a few points that require more discussion.

**Comments:**

- The manuscript does not adequately contextualize the work with regards to other wake turbulence models. While the work by Ishihara and Qian is mentioned, there are no comparisons to the proposed model and so advantages of the current approach cannot be fully assessed. Moreover, the work by Crespo and Hernandez [1, 2], which remains the prevailing wake-added turbulence model used in the wind energy industry and research communities is not mentioned at all.
- The two parts of the manuscript overlap a great deal. Both parts contain a description of the mean flow, and of components of the turbulence field (meandering and fixed frames of reference). I recommend either combining and consolidating the work into a single article or working to distinguish the content in each.
- The notation $\hat{\cdot}$ is not defined in the current work and requires readers to look at Part I. It's also not clear why the notation must also require subscripts to distinguish between quantities in the meandering or fixed frames if the hat notation does the same job. In some terms, the authors use subscripts for meandering frame and for a reference velocity field, which is the undisturbed ABL. This seems like a contradiction. What is meandering in the reference field?
- Many equations are repeated between the two Parts of the article. Equations 6 and 7 contain many terms that are not given enough description or physical interpretation in Part II. Please add a brief description for each term.
- Equation 7 defines several forms of shorthand for some terms (e.g., $k_m = (\mathrm{III})$). Why are multiple names used?

- The definition of turbulence used by the authors appears to arise from decomposing TKE in a fixed frame into the meandering frame of reference. The turbulence model is only coupled to the velocity model through the decay function $C(x)$ given in Eq. 2. In reality, the turbulence field arises from mean shear gradients, solid body interactions and boundary layers. Is this model sufficient to describe changes in turbulence due to changes in the mean momentum deficit and wake morphology? Does the velocity model depend in any way on the turbulence field?
- On lines 66 and 67, the authors state that the mechanisms for wake meandering and wake expansion are treated independently. This strong assumption is not likely to hold in all cases. Can the authors offer more reasoning for this decision? What are the consequences of treating the mechanisms separately? Are the cross terms in Eq. 7 responsible for the coupling of these mechanisms?
- Neither the velocity model nor the turbulence model make use of the stable simulation discussed in Part I of the paper. The authors state that the stable case is not modeled because veer is not described in the current formulation, but that it could be in the future. This is arguably one of the most important cases to model as it leads to the greatest wind plant wake losses, and should be included in the current work.
- Stability is not described in the models. The changes introduced by stability must then come from the dizzying array of standard deviations listed in the models. Is this level of empiricism a step forward from existing wake velocity and turbulence models?
- The authors state that "only atmospheric parameter that seems to influence $\Delta TI_{MF}$ is the shear...". This is not the only boundary condition that should be considered. Even in the neutral case, the roughness length will determine the velocity and turbulence profile, the characteristic length scales of inflow turbulence, correlation lengths for meandering, etc. In stable and unstable cases, the surface heat flux will be important to fully describe the sources/sinks of momentum and turbulent kinetic energy. The authors must discuss limitations in the modeling approach and consequences in the final predictions. These sources of uncertainty may be the limiting factor of the model in the end.
- The wake-added turbulence shown in Figures 3 and 14 does not appear to be fully converged. What is the uncertainty associated with developing the model with poorly converged statistics?
- Eq. 15 is shown in the text and can be removed.
- Are Eqs. 15–17 used to infer a value of $l_m$? If so, what are the consequences of neglecting so many non-zero gradients in the rate-of-strain tensor? What about using a log-layer estimate of $u_0$? Finally, is a single value of $l_m$ used for the full model everywhere in the wake? There are challenges and limitations with this approach discussed by Iungo [3, 4] and Martínez [5, 6]. Please discuss the model and assumptions in the context of previous work.
- The notation in Eq. 19 needs to be changed from $e$ to $\exp$ to be consistent with the rest of the manuscript.
- How will the models used in the current work be validated? There are not many sources of utility-scale wind turbine wake turbulence available for research.

**References**

[1]  Antonio Crespo, J Herna, et al. "Turbulence characteristics in wind-turbine wakes". In: *Journal of wind engineering and industrial aerodynamics* 61.1 (1996), pp. 71–85.

[2]  Nicholas Hamilton et al. "Comparison of modular analytical wake models to the Lillgrund wind plant". In: *Journal of Renewable and Sustainable Energy* 12.5 (2020), p. 053311.

[3]  Francesco Viola et al. "Prediction of the hub vortex instability in a wind turbine wake: stability analysis with eddy-viscosity models calibrated on wind tunnel data". In: *Journal of Fluid Mechanics* 750 (2014).

[4]  Giacomo V Iungo et al. "Data-driven RANS for simulations of large wind farms". In: *Journal of Physics: Conference Series*. Vol. 625. 1. IOP Publishing. 2015, p. 012025.

[5]  Luis A Martínez-Tossas et al. "The aerodynamics of the curled wake: a simplified model in view of flow control". In: *Wind Energy Science* 4.1 (2019), pp. 127–138.

[6]  Luis A Martínez-Tossas et al. "The curled wake model: a three-dimensional and extremely fast steady-state wake solver for wind plant flows". In: *Wind Energy Science* 6.2 (2021), pp. 555–570.

---

## Author Response (AR1)

Dear editor and reviewers, thank you for taking the time to examine our work. Several weak points were pointed out, which we hope to have comprehensively answered in the final version. Please find our answer (in bold blue) to the reviewers comments (in black).

**RC1**

The authors developed an analytical model for the velocity and turbulence in the wake of a wind turbine taking meandering into account. The overall topic is an important one, as there is a need for improving engineering models. However, some of the assumptions made in the derivation of the model seem to be poor choices, for example, using the Gaussian shape hypothesis of the velocity deficit in the near wake and the Gaussian distribution of the wake center in the far wake. Thus, excluding the veer impact is a big question even for the unstable case. The authors keep giving pieces of advice on how to develop the current model in many places, which provides a negative impression of the current work. Although the results are encouraging, and my overall impression of this manuscript is positive, the authors should do some revisions to re-evaluate their work objectively.

1. The authors need to show a comparison between the new model with known models to evaluate the work.
   **The authors think that a comparison with another model would be confusing because our model has not been calibrated and its parameters have been optimized to get the best results compared to Meso-NH. A true comparison with other models will be possible once our model is calibrated.**

[Figure]

**Although, we added results from the model of Ishihara and Qian to show the capability of our model to give different shapes for the unstable and neutral cases whereas the IQ2018 gives similar results. We added the following lines:**

*Figure 1 Modified Figure 12*

*"In green is also plotted the model of Ishihara and Qian (2018), denoted IQ2018 hereafter. The results from IQ2018 are obtained from the values of CT, Tix and Kx upstream of the turbine. It should be noted that the comparison is not very fair because our model has not been calibrated and thus does not depend on calibration like IQ2018. We can note that the IQ2018 model gives fairly good results for vertical profiles, due to the correction near the ground proposed by the authors. However, for the IQ2018 profile to show a peak at the top tip, it needs to also show a double peak for the y profile (see Fig.12 at x/D=8), a phenomenon that is not observed in the LES and that is not necessarily seen in our model due to the definition of $\sigma$ and $\sigma_f$ in the two directions."*

[Figure]

*Figure 2 Modified figure 13*

*"The unstable case shows the main shortcoming of the IQ2018 model and the added value of our model. Besides the upstream turbulence profiles, the inflow conditions used in the IQ2018 are very similar between the neutral and unstable cases. Consequently, the green profiles are alike in Figs. 12 and 13 whereas the stronger meandering in the unstable case leads to a Gaussian-like turbulence profile, even in the vertical direction. The*

*maximum turbulence is thus no longer located at the top tip but rather at hub height. This property is well-predicted by our model whereas the IQ2018 model, which does not take meandering into account, predicts quasi-identical behaviours in the neutral and unstable cases."*

2. The authors assumed that "It appears that the results are much better in the neutral case (in green) than in the unstable case (in red). This is likely due to the higher meandering in the unstable case, which would require a higher number of data to reach a converged PDF." How can one trust the result if we do not have converged statistics, especially for turbulence?
**This is indeed a weakness of our work, that we acknowledged only during the post-processing. At first, it was expected that 40-minutes statistics would be sufficient to have converged data in the unstable case, which does not seem to be the case. Our results must thus be taken with care, and this is one of the reasons why we did not calibrate the model on the current dataset. However, it was interesting to include this unstable case because with a similar Ct and TI at hub height than the neutral case, it gives significantly different results in the wake, which is the reason why we developed the present model.**

**There is also a certain utility for the scientific community to show that even 40-minutes averages are not sufficient to get converged data in unstable ABL. For a further study to calibrate the model, this must be taken into account.**

3. The authors need to justify that the impact of the veers is negligible in the unstable case.
**This is simply an observation of the wind direction profile upstream the turbine, which happens to be almost constant with height over the rotor-swept area. The profiles are plotted in the figure below: between the ground and 90 metres (i.e. more than 2D above hub height) above the ground the gradient of wind direction in the neutral case is about 1.25°. In the unstable case, it is even less.**

[Figure]

**We thus added the following remark in line 105:**

*"the veer upstream the turbine is negligible: respectively, the difference between the maximum and minimum wind direction between the ground and 90 m above the ground is of $1.25\degree$ and $0.5\degree$."*

4. The authors need to check if possible the impact of the turbulent intensity on the model, since the structure of the turbulence depends on that.
**The authors are not sure to understand the reviewer's comment. If it is about the impact of the inflow's TI on the LES data, we could not do so with the present dataset since we had only two cases with similar TI levels.**
**If it is about how the inflow TI will be included in the model, it is planned for the future. Basically, so far the model is not calibrated, i.e. the parameters are not expressed as a function of the inflow. This will be done in the future with a larger dataset containing more cases. Once enough data are gathered, we will be able to write $\sigma_y$, $\sigma_z$ and $\sigma_{fy}$, $\sigma_{fz}$ as a function of the inflow TI. For $\sigma_y$, $\sigma_z$ we expect a dependency on $I_x$ as in [8] for instance. For $\sigma_{fy}$, $\sigma_{fz}$ we expect to write them as a function of $I_y$ and $I_z$, respectively, but this is only a supposition and must be confirmed by future works.**

**RC2**

Summary: The manuscript entitled "Breakdown of the velocity and turbulence in the wake of a wind turbine - Part 2: Analytical modeling" endeavors to describe the turbulent velocity field in a wind turbine wake by accounting for energy in both the meandering and fixed frames of reference. The mathematical development is thorough and detailed, even if the presentation is difficult to follow at times. The model development takes cues from some well known approaches in fundamental turbulence (e.g., the Boussinesq Hypothesis) and is almost entirely analytical, leaving very few constants to be tuned empirically. For the most part, the manuscript is well written and clear, although there are a few points that require more discussion.

Comments:

• The manuscript does not adequately contextualize the work with regards to other wake turbulence models. While the work by Ishihara and Qian is mentioned, there are no comparisons to the proposed model and so advantages of the current approach cannot be fully assessed. Moreover, the work by Crespo and Hernandez [1, 2], which remains the prevailing wake-added turbulence model used in the wind energy industry and research communities is not mentioned at all.

**Thank you for the advice, we added a reference to the work of Crespo as well as the more recent work of Stein:**

*"For the turbulent kinetic energy (TKE), it is common to model only the maximum value of added turbulence which can be computed with the Crespo model (Crespo and Hernandez, 1996) or the Frandsen model (Frandsen, 2007) as in the IEC 61400-1 standard. Their approach is mainly empirical and can be extended to describe the whole profile of turbulence instead of the maximum value alone (Ishihara and Qian, 2018). More recently, a physically-based model for each Reynolds' stress component has been proposed based on self-similarity (Stein and Kaltenbach, 2019)."*

**And at the end of the introduction for more context about our work:**

*"We will show that due to meandering, the turbulence in the wake no longer respects self-similarity and that another parameter is needed to yield correct shape functions in the wake."*

**According to the first remark of RC1, we added a comparison to the work of IQ2018 in the results.**

• The two parts of the manuscript overlap a great deal. Both parts contain a description of the mean flow, and of components of the turbulence field (meandering and fixed frames of reference). I recommend either combining and consolidating the work into a single article or working to distinguish the content in each.

**The authors chose to split the article into two parts because there was two distinct matters. The Part I focuses on the development of the turbulence breakdown and the quantification of other terms. This first part is not limited to steady analytical models and shows some underlying assumptions of DWM-type models, even when they are used in unsteady mode because they neglect cross-terms. The second part is a new steady analytical model. Readers may thus be interested in one part and not the other. We think it is pertinent to submit two papers to avoid needless reading efforts for those that are not interested by the whole work (which would be about 40 pages if submitted as a whole).**

**We thought it pertinent to describe some things twice such as mean flow and breakdowns so the two parts are nearly independent and so the reader must not read both.**

• The notation ˆ is not defined in the current work and requires readers to look at Part I. It's also not clear why the notation must also require subscripts to distinguish between quantities in the meandering or fixed frames if the hat notation does the

same job. In some terms, the authors use subscripts for meandering frame and for a reference velocity field, which is the undisturbed ABL. This seems like a contradiction. What is meandering in the reference field?

**The introduction of the notation has been modified to be consistent with the companion paper (line 55).**

**There is indeed no meandering in the reference simulation because there is no wake at all. However, if we define $y_c(t)$ and $z_c(t)$ equal to those in the simulation with the turbine, we can define a meandering frame for this reference simulation.**

**This approach has been decided so the following equation holds for the LES dataset:**

$$\Delta k_{FF} = k_{FF} - k_{FF,ref} = \widehat{k_{MF}} - \widehat{k_{MF,ref}} = \overline{k_{MF} - \widehat{k_{MF,ref}}} = \overline{\Delta k_{MF}}$$

• Many equations are repeated between the two Parts of the article. Equations 6 and 7 contain many terms that are not given enough description or physical interpretation in Part II. Please add a brief description for each term.
Equation 7 defines several forms of shorthand for some terms (e.g., km = (III)). Why are multiple names used?

**Two paragraphs have been added to answer these remarks:**

**"The different terms, noted from (I) to (VII), are separated into pure- and cross-terms. They are thoroughly described and quantified in the companion paper. The term (I) is the convolution of UM F with fc. It is a pure mean velocity term: it is null only if the mean velocity is null. Conversely, the term (II) is a cross-term because it can be equal to 0 either if there is no meandering (x = x) or if there is no turbulence in the MFOR (U ' M F = 0).**

**The term (III), also written km in the following to be consistent with notation from Keck et al. (2013) and Conti et al (2021), is the turbulence purely induced by meandering: in the case of a meandering steady wake i.e. U ' M F = 0, Eq. 7 reduces to this term only. The term (IV) is the rotor added turbulence, which is also written ka for consistency with other works. It is the turbulence purely induced by the rotor: in absence of meandering (x = x), the equation reduces to this term only, also written ka in the following. Term (V) is the covariance of ˆUM F and ˆU ' M F , term (VI) can be viewed as the varying part of the MFOR turbulence and term (VII) is the square of the term (II). It is a pure dissipation term as it is always negative. Like the term (II), they are cross-terms since they are equal to zero if either the turbulence in the MFOR or the meandering is null."**

• The definition of turbulence used by the authors appears to arise from decomposing TKE in a fixed frame into the meandering frame of reference. The

turbulence model is only coupled to the velocity model through the decay function C(x) given in Eq. 2. In reality, the turbulence field arises from mean shear gradients, solid body interactions and boundary layers. Is this model sufficient to describe changes in turbulence due to changes in the mean momentum deficit and wake morphology? Does the velocity model depend in any way on the turbulence field?

**In our model, the turbulence (in the MFOR) is assumed to entirely come from velocity gradients of U in the y and z directions (cf 2$^{nd}$ line of Eq. 19). This assumption is of course questionable but is convenient to develop our model. The wake is here assumed to take a Gaussian shape, so the morphology of the wake is given by parameters $\sigma_y$ and $\sigma_z$. A Super- or Double-Gaussian shape model could be developed in the future to improve this morphology in the near wake, hence adding other parameters, but it was not in the scope of this work.**

**Conversely, the velocity model is often assumed to depend on the turbulence field through a dependency of $\sigma$ on $k_x$ (or its TI form). But this dependency does not vary with $y$ and $z$.**

• On lines 66 and 67, the authors state that the mechanisms for wake meandering and wake expansion are treated independently. This strong assumption is not likely to hold in all cases. Can the authors offer more reasoning for this decision? What are the consequences of treating the mechanisms separately? Are the cross terms in Eq. 7 responsible for the coupling of these mechanisms?

**A realistic wake is simultaneously expanding and meandering. In most analytical models it is assumed that on average these two phenomena can be treated as an expansion, but here they are treated as two different phenomena, as they are in reality. The level of assumption is thus lower in our work than in most models.**

**This approach is not new [9,10,11] but one of the originalities of our work is that Eqs. 6 and 7 have been developed to show that indeed, these two phenomena are not entirely independent. However, the quantification of the different terms in the companion paper has shown that the cross terms, responsible for the interaction between expansion and meandering as you said, are negligible as a first approximation. The induced error can be seen by the differences between the blue and black curves in Figs. 7, 8, 13 and 14, which is often less important than the other assumptions of the model, that lead to the differences between orange and blue curves in the same graphs.**

**To clarify, we added (line 80):**

***"It is common in wake modelling to assume that meandering can be entirely accounted for by increasing the wake expansion. In the present model, these***

*phenomena are modelled separately, but it will be assumed that they do not interact. This is equivalent to neglecting cross-terms in Eqs. 6 and 7 which have been shown to take consistently smaller values than pure-terms in the companion paper. In the future though, modelling these cross-terms might be necessary to improve the results."*

**And in the conclusion;**

*"As shown in Figs. 7, 8, 13 and 14 the error induced by neglecting cross-terms (between black and blue curves) is lower than the error of the model itself (between blue and orange curves) but modelling these terms could improve the results, in particular in the vertical direction."*

• Neither the velocity model nor the turbulence model make use of the stable simulation discussed in Part I of the paper. The authors state that the stable case is not modeled because veer is not described in the current formulation, but that it could be in the future. This is arguably one of the most important cases to model as it leads to the greatest wind plant wake losses and should be included in the current work.

**A model for the stable case has not been proposed, mainly for two reasons:**

- **The model of wakes under veer, as proposed by Abkar [1] could indeed be used. However, it uses a tangent function, which depends on z. Therefore, derivations such as Eqs. 19, 28 or 30 would become much harder to resolve, and may not even have an analytical solution.**
- **Moreover, in stably stratified ABL, the meandering becomes negligible, and the meandering turbulence as well (cf companion paper). For such a case, our approach does not bring a lot, and one could simply use a model that does not predict meandering explicitly since, meandering is negligible.**

**We thus modified line 101 as:**

*"In the companion paper, three cases of stability were simulated but the stable case has been discarded for this paper. Indeed, the veer present in such a case can be modelled as in Abkar et al (2018) but would significantly complicate the present derivations. Moreover, meandering and meandering turbulence are negligible in a stably stratified ABL (see companion paper) and thus there is little interest in using the approach presented herein."*

• Stability is not described in the models. The changes introduced by stability must then come from the dizzying array of standard deviations listed in the models. Is this level of empiricism a step forward from existing wake velocity and turbulence models?

**Please note that we did not propose any calibration of the model's parameters yet. Instead, the parameters have been deduced directly from the simulations, according to Sect 3.4. Writing a relation between the model parameters and inflow conditions is thus not in the scope of this work.**

**In the future it will be possible to write $\sigma_{fy} = f(\zeta)$ for instance. However, we do not think that it is a good idea: according to the work of Du et al. [2], buoyancy itself does not play any role in the recovery of wind turbine wakes. Thus, it is possible that the impact of stability on recovery is only due to modification of the std of lateral and vertical velocities. We may instead model $\sigma_{fy} = f(I_y)$ in the future. Maybe we will need $\sigma_{fy} = f(I_y, \Lambda_y)$ where $\Lambda_y$ is the integral length scale because as we showed in [3] the integral length scale is an important parameter for meandering. These parameters could be related to $\zeta$ but we think it is important to relate a phenomenon to its direct cause, i.e. std of velocities.**

• The authors state that "only atmospheric parameter that seems to influence ΔT IMF is the shear…". This is not the only boundary condition that should be considered. Even in the neutral case, the roughness length will determine the velocity and turbulence profile, the characteristic length scales of inflow turbulence, correlation lengths for meandering, etc. In stable and unstable cases, the surface heat flux will be important to fully describe the sources/sinks of momentum and turbulent kinetic energy. The authors must discuss limitations in the modeling approach and consequences in the final predictions. These sources of uncertainty may be the limiting factor of the model in the end.

**As shown in [2], the heat fluxes in the wake are negligible in the mean kinetic energy budget. It may take a more important role in the TKE budget but we expect the transport and convection terms to be preponderating. This is confirmed by the fact that our turbulence field in the MFOR is very similar between the neutral and unstable cases. The role of integral length scales has been studied in [3].**

**We agree that our statement is rather bold, especially since it is based on only two cases with the same level of surface roughness and thrust coefficients. We thus replaced**

*"Similarly to the velocity deficit, the added turbulence field in the MFOR is very similar between the two cases of stability, indicating that the turbulence added by the rotor depends more on the operating conditions (CT , tip-speed ratio) than on the atmospheric conditions (velocity at hub height, atmospheric stability...). The only atmospheric parameter that seems to influence ΔT IM F is the shear (in particular in the neutral case), which breaks the symmetry of the wake as it travels downstream."*

**With**

*" Similarly to the velocity deficit, the added turbulence field in the MFOR is very alike between the two cases of stability. Atmospheric stability and hub-height velocity are thus not parameters of the added turbulence in the MFOR, as long as sufficiently large turbulent structures are present in the inflow (Jézéquel et al., 2022). Instead, shear has a clear effect in the neutral case, by breaking the symmetry of the wake as it travels downstream. Other parameters, such as thrust coefficient or roughness length, may impact ΔT IM F but are here constant among the two cases so more work is needed to estimate their impact."*

• The wake-added turbulence shown in Figures 3 and 14 does not appear to be fully converged. What is the uncertainty associated with developing the model with poorly converged statistics?

**Please refer to our answer to question 2 of RC1. We acknowledge that it is a weakness of our study but think that the unstable case is still worth to be shared with the scientific community. For further works, in particular calibration of the model, this will be more closely looked upon.**

• Eq. 15 is shown in the text and can be removed.

**Thank you for this remark, it has been removed in the text instead.**

• Are Eqs. 15–17 used to infer a value of lm? If so, what are the consequences of neglecting so many non-zero gradients in the rate-of-strain tensor? What about using a log-layer estimate of u0? Finally, is a single value of lm used for the full model everywhere in the wake? There are challenges and limitations with this approach discussed by Iungo [3, 4] and Mart´ınez [5, 6]. Please discuss the model and assumptions in the context of previous work.

**For the present work, the mixing length has been deduced through an optimization process (cf part 3.4). This has been done because at first, using**

the mixing length formulation proposed by Iungo et al., 2017 [12] (Eq. 15) failed. Calibration of this mixing length is expected to be a challenge.

Neglecting some of the gradients has been done based on the work [12] (fig 2, we see that most of the strain rate tensor comes from $\partial U_r / \partial x$), which we supposed to be also true in the MFOR.

Thank you for suggesting a vertical-dependent value as in the works of Martinez-Tossas. There are indeed many models for the mixing length available in the literature, like Prandtl ($l_m = \kappa z$), Blackadar [4] (i.e. the one used by Martinez-Tossas) or Grisogono and Belušić [5]. However, this choice has not been made for the following reasons:

- All these mixing lengths are developed to model the ABL at large scales, usually much larger than the typical size of a wind turbine wake. We are here not trying to model the same phenomena and thus should not necessarily use the same parametrization.
- Mixing lengths of Prandtl or Blackadar would result in a mixing length solely dependent on $z$, whereas we want something that is dependent on $x$ as well to be consistent with the previous work of[12]. This is confirmed by our Fig. 6 where $l_m$ found with an optimisation algorithm at each $x$ reveals a value that depends strongly on $x$.
- A mixing length like that in [5] i.e. $l_m = min\left(\frac{A_1\sqrt{k}}{N}, \frac{A_2\sqrt{k}}{S}\right)$ where N and S are the Brunt-Vaisala frequency and the strain rate norm could be pertinent but once inserted in our equation it would simplify $k$, that would no longer be present in the equation which we obviously don't want.
- One could argue that a value of $l_m$ dependent on $z$ would improve the dependency of the model on $z$. That may be true but to our interpretation, this would be due to an error compensation because the weak asymmetry of $k_{x,MF,am}$ compared to the LES value is due to our simplifications on shear. The $z$-dependent models like [4] and [5] are developed to model the tendency of atmospheric eddies in the atmosphere to grow in size as they elevate above the ground. To our interpretation, there is no reason that it is the case inside a wake, especially in the MFOR. We recall that our mixing length is only used for the wake in the MFOR so we do not need to have a realistic mixing length for the atmosphere.

Assuming $l_m = f(x)$ only, one could relate $l_m$ to the wake width $\sigma(x) = \sqrt{\sigma_y \sigma_z}$ as in Keck et al (2012) [6], but this approach did not give good results. That is why an optimisation method has been chosen to find the values of $l_m$ for this work.

**Following your recommendations, this contextualization has been summed up after Eq. 19:**

*" Computing the mixing length in Eq. 19 is a challenge that has not been answered yet in this work. Formulations that depend on the vertical coordinate (like the Prandtl mixing length lm = κz or the modified version of Blackadar (1962)) are not appropriate herein because they would result in a value constant with x whereas a previous work showed the opposite in a wind turbine wake (Iungo et al., 2017). Local formulations such as Grisogono and Belušĭ c (2008) could also be used but would increase the complexity of the model and for this particular case would lead to the simplification of kx,M F,am, which we want to avoid. Moreover, these formulations have been developed for the ABL whereas we are looking for a mixing length to apply to the wake in the MFOR, which is not driven by the same phenomena.*

*It has thus been decided to use a mixing length that only depends on the streamwise direction lm(x). Two mixing length values proposed in the literature have been tried (Keck et al., 2012; Iungo et al., 2017) without success. However, the authors think that this type of formulation is more appropriate than those aforementioned but needs some modifications to fit our model. A proper formulation of the mixing length will be proposed in further works, but for the present work the value of lm at each position downstream is deduced through an optimisation algorithm (see Sect. 3.4)."*

• The notation in Eq. 19 needs to be changed from e to exp to be consistent with the rest of the manuscript.

**Thank you for noticing this typo, it has been corrected.**

• How will the models used in the current work be validated? There are not many sources of utility-scale wind turbine wake turbulence available for research.

**We plan to perform new LES simulations to complete this work with proper calibration. Validating the models against utility-scale turbines might indeed be an issue. However, some facilities are equipped with high-frequency scanning lidars that allow computing turbulence in the wake of a real turbine (and even in the MFOR such as [7]). These are promising and we are looking forward to get access to more data of this type.**

**Moreover, some wind tunnels can simulate non-neutral ABLs and perform turbulence measurements in the wake of wind turbines. In absence of data from utility-scale wind turbines, we may use such data for validation.**

**References**

[1] Abkar, M., Sørensen, J., and Porté-Agel, F.: An Analytical Model for the Effect of Vertical Wind Veer on Wind Turbine Wakes, Energies, 11,1838, https://doi.org/10.3390/en11071838, 2018

[2] Du, B.; Ge, M.; Zeng, C.; Cui, G. & Liu, Y. Influence of atmospheric stability on wind turbine wakes with a certain hub-height turbulence intensity Physics of Fluids, 2021

[3] Jézéquel, E.; Blondel, F. & Masson, V. Analysis of wake properties and meandering under different cases of atmospheric stability: a large eddy simulation study Journal of Physics: Conference Series, IOP Publishing, 2022, 2265, 022067

[4] Blackadar, A. K. The vertical distribution of wind and turbulent exchange in a neutral atmosphere Journal of Geophysical Research (1896-1977), 1962, 67, 3095-3102

[5] Grisogono, B. & Belušić, D. Improving mixing length-scale for stable boundary layers Quarterly Journal of the Royal Meteorological Society, Wiley, 2008, 134, 2185-2192

[6] Keck, R.-E.; Veldkamp, D.; Madsen, H. A. & Larsen, G. Implementation of a Mixing Length Turbulence Formulation Into the Dynamic Wake Meandering Model Journal of Solar Energy Engineering, 2012, 134

[7] Larsen, G.; Pedersen, A.; Hansen, K.; Larsen, T.; Courtney, M. & Sjöholm, M. Full-scale 3D remote sensing of wake turbulence - a taster J Phys : Conf Ser, IOP Publishing, 2019, 1256, 012001

[8] Fuertes, F. C.; Markfort, C. & Porté-Agel, F.Wind Turbine Wake Characterization with Nacelle-Mounted Wind Lidars for Analytical Wake Model Validation Remote Sensing, 2018, 10, 668

[9] Keck, R-E; Maré, M.; Churchfield, M. J; Lee, S. Larsen, G. & Madsen, H. A. Two improvements to the dynamic wake meandering model: including the effects of atmospheric shear on wake turbulence and incorporating turbulence build-up in a row of wind turbines; Wind Energy, 2013

[10] Braunbehrens, R. & Segalini, A. A statistical model for wake meandering behind wind turbines Journal of Wind Engineering and Industrial Aerodynamics, 2019, 193, 103954

[11] Conti, D.; Dimitov, N.; Peña, A. & Herges,T.  Probabilistic estimation of the Dynamic Wake Meandering model parameters using Spinner-Lidar derived wake characteristics, Wind Energy Science, 2021

[12] Iungo, G, V.; Santhanagopalan, V.; Ciri, U.; Viola, F.; Zhao, L.; Rotea, M.A., & Leonardi, S. Parabolic RANS solver for low-computational-cost simulations of wind turbine wales ; Wind Energy, Wiley, 2017

---

## Referee Report (RR1)

**REVIEW OF WES-2022-47.R1**

*Breakdown of the velocity and turbulence in the wake of a wind turbine - Part 2: Analytical modeling.*

*authors:*
Erwan Jézéquel
Frederic Blondel
Valery Masson

**Summary:**

The manuscript entitled "Breakdown of the velocity and turbulence in the wake of a wind turbine - Part 2: Analytical modeling" endeavors to describe the turbulent velocity field in a wind turbine wake by accounting for energy in both the meandering and fixed frames of reference. Many of the comments that arose during the previous round of reviews were addressed by the authors, and I'd like to thank them for their frank and direct responses. In all, the manuscript describes important work deriving an analytical expression for wind turbine wake turbulence. However, because the work ends with a theoretical description, it remains unclear how effective the model will be in application for wind plant simulation, prediction, controls, design, etc. Without calibrating or validating the proposed model, training over a broader range of atmospheric conditions, error analysis, uncertainty estimation, and detailed comparison to existing models, the proposed work is incomplete and will not be likely to have the intended impact on the field of wind engineering.

**Comments:**

- The authors state in the opening sentence of the abstract that the novelty and benefit of the proposed model is that, "the expansion and meandering of the wake can be independently calibrated." However, no attempt is made to complete this step, and only a limited range of large eddy simulations were used to deduce model parameter values. This is a necessary step before the model can be validated and its range of application understood.

- The authors stated in their response to the previous comments that model parameters should be related to underlying causes (i.e., the standard deviations of velocities) rather than stability metrics. However variability in the velocity field is in fact a product of both mechanical and thermally driven turbulence. The results of the reference pointed out by the authors [1] concluded that, "With the same turbulence intensity, atmospheric stability can significantly change the turbulent kinetic energy distribution in the three spatial directions." This can only emphasize the importance of accounting for buoyancy in the model.

- The authors indicate that comparison to existing wake-added turbulence models would be confusing without calibration. I agree with this point in that results of such a comparison would be difficult to interpret, but I see it as another reason to pursue calibration data, rather than as a reason to omit model comparisons, error estimates, and uncertainty analysis.

**References**

[1] Bowen Du et al. "Influence of atmospheric stability on wind-turbine wakes with a certain hub-height turbulence intensity". In: *Physics of Fluids* 33.5 (2021), p. 055111.

---

## Author Response (AR2)

Both reviewers underline that the model is not calibrated nor validated. The authors undertake that in this current form, the model is more a proof of concept than a fully detailed model. We however wanted to publish our results at this point because we think it is still of interest to the scientific community. There are two main reasons for this:

1. This model is based on a new theoretical approach (modelling in the MFOR and then solving convolution products), and thus has added value independently from the resulting formula of the model (Eqs. 35 and 36). This approach can be re-used by other research teams which would like to develop their own models based on the same methodology, but with other shape functions in the MFOR. For instance, deriving the model with a super-gaussian function.
The two LES cases are only used here to choose appropriate shape functions in the MFOR, verify that our results are consistent, and show the possibilities offered by our model to give different results with same $C_T$ and $TI_x$.

2. Since we plan to perform an in-depth calibration, we cannot perform it on only two cases. Thus, we need results from new simulations, with many atmospheric and operating conditions. Moreover, we would like to validate the model against in-situ data if possible, or at least wind tunnel measurements or another set of LES. Depending on the variables chosen to calibrate the model, such dataset might be hard to find, or might even demand us to ask for experimental results with our partners. In other words, we think that a proper validation/calibration of our model would be sufficient for another article, or at least a conference proceeding.

Despite this, we have some ideas for the calibration, for instance we got good results with the relation $\sigma_{fy}(x) = I_y^D x$, where $I_y^D$ is the upstream, lateral turbulent intensity averaged over the rotor disk (and similarly for $\sigma_{fz}$). However, this is only observed on the three cases and needs to be generalised on more data before publication. For $\sigma_y$ and $\sigma_z$, we expect these parameters to be a function of $C_T$ and maybe $I_x$. The mixing length $l_m$ might require more in-depth studies.

We added in section 3.4 (before the bullet points):
*The model's parameters are not known a-priori: to have a usable model, it is planned to link them to the upstream flow quantities. In particular, a dependency of $\sigma_{fy}$ on the lateral turbulence intensities and the integral length scale has been observed. However, this is only observed on the present cases and needs to be generalised on more data before publication. Due to the small amount of data at disposal, the present work does not aim at calibrating properly the modelled terms but simply to show that a simple shape function can already lead to a rather good approximation. Therefore, the values of the parameters are here directly deduced from the LES field:*

We added in the introduction:
*The presented model is not calibrated herein. Nevertheless, the added value of this work is to propose a new framework that can be used with different shape functions in the MFOR to propose other models for turbulence.*
And in the conclusion:

*Finally, the presented model is a proof of concept and a calibration (i.e. relating different parameters σ, σf and lm to the inflow conditions) under different atmospheric conditions is necessary before it can be used.*

And in the abstract:

*This model is a proof a concept that shows a methodology where one can calibrate a model in the fixed frame of reference (FFOR) with the use of shape functions chosen in the moving frame of reference (MFOR), and therefore modelling physically the added turbulence*

**Reviewer 1**

This reviewer thanks the authors for answering the comments. However, the authors did not provide enough details regarding critical comments. For example:

(1) no clear comparison with other models is provided because their model is uncalibrated.
   Please report to our answer at the beginning of the document.

(2) presenting the results without converging statistics (unstable case).
   This remark has also been done by reviewer 1 of the companion paper (wes2022-46). A statistical analysis is proposed in our next submission, however, we did not think it would be pertinent to write it on both papers so it will not appear in this paper. The following text has been added in the methodology section:
   *Moreover, the duration of the simulation is set to 80, 40 and 10 minutes for the neutral, unstable and stable cases, respectively. An analysis of the statistical convergence of our datasets is proposed in appendix of the companion paper. Overall, it concluded that increasing the duration of simulation for the unstable case would improve the reliability of the simulations. Nevertheless the convergence of the results is assumed to be sufficient since here it is aimed to propose a proof a concept and not a fully developed model.*

(3) There is no clear answer regarding the impact of the TI on the result because there had only two cases with similar TI levels.
   This indeed will need to be addressed in future work when calibrating the model. Simulations with different turbulence at hub height, and possibly independently different atmospheric stabilities, should be performed.

Although these points are critical to validate any introduced model, my overall impression of this manuscript is still positive. Therefore, the authors need to clarify these points in the abstract and conclusion, before recommending the work for publication.

**Reviewer 2**

The manuscript entitled "Breakdown of the velocity and turbulence in the wake of a wind turbine - Part 2: Analytical modeling" endeavors to describe the turbulent velocity field in a wind turbine wake by accounting for energy in both the meandering and fixed frames of reference. Many of the comments that arose during the previous round of reviews were addressed by the authors, and I'd like to thank them for their frank and direct responses. In all, the manuscript describes important work deriving an analytical expression for wind turbine wake turbulence. However, because the work

ends with a theoretical description, it remains unclear how effective the model will be in application for wind plant simulation, prediction, controls, design, etc. Without calibrating or validating the proposed model, training over a broader range of atmospheric conditions, error analysis, uncertainty estimation, and detailed comparison to existing models, the proposed work is incomplete and will not be likely to have the intended impact on the field of wind engineering.

**Comments:**

- The authors state in the opening sentence of the abstract that the novelty and benefit of the proposed model is that, "the expansion and meandering of the wake can be independently calibrated." However, no attempt is made to complete this step, and only a limited range of large eddy simulations were used to deduce model parameter values. This is a necessary step before the model can be validated and its range of application understood.
*This sentence has been changed to **"the expansion and the meandering are taken into account independently"** in order to avoid misleading the reader.*

- The authors stated in their response to the previous comments that model parameters should be related to underlying causes (i.e., the standard deviations of velocities) rather than stability metrics. However variability in the velocity field is in fact a product of both mechanical and thermally driven turbulence. The results of the reference pointed out by the authors [1] concluded that, "With the same turbulence intensity, atmospheric stability can significantly change the turbulent kinetic energy distribution in the three spatial directions." This can only emphasize the importance of accounting for buoyancy in the model.
*The authors agree that atmospheric stability is indeed of primary importance for predicting the overall wake recovery. As shown in your citation, this is allegedly attributed in reference [1] to modifications of TKE distribution in the three spatial directions. Moreover, in [1], a mean kinetic energy budget (page 6) shows that the buoyancy term has a negligible impact on wake recovery. It thus seems that thermal effects modify the wake recovery because it modifies the distribution of TKE and the eddies' size, but not because of the buoyancy itself, and thus this buoyancy can be neglected. We did not verify this assumption, but we showed in a previous work [2] that between the neutral and unstable cases, the flow is almost identical in the MFOR.*
*It may be needed to write $\sigma_{fy}$ and $\sigma_{fz}$ as a function of the stability, but if this can be avoided and only written as a function of the directional TKE, it will be preferred because such data will be easier to acquire. Maybe writing it as a function of the integral time scale will be needed as in the Taylor diffusion theory [3]. We think that all these questions about calibration are worth addressing as an entire paper and not only a quick note, and that is also why we wanted to address it later.*

- The authors indicate that comparison to existing wake-added turbulence models would be confusing without calibration. I agree with this point in that results of such a comparison would be difficult to interpret, but I see it as another reason to pursue calibration data, rather than as a reason to omit model comparisons, error estimates, and uncertainty analysis.
*Please report to our answer at the beginning of the document.*

[1] Bowen Du et al. *"Influence of atmospheric stability on wind-turbine wakes with a certain hub-height turbulence intensity".* In: Physics of Fluids 33.5 (2021), p. 055111.

[2] Jézéquel, E.; Blondel, F. & Masson, V. *"Analysis of wake properties and meandering under different cases of atmospheric stability: a large eddy simulation study"* Journal of Physics: Conference Series, IOP Publishing, 2022, 2265, 022067

[3] Cheng, W.-C. & Porté-Agel, F. *"A Simple Physically-Based Model for Wind-Turbine Wake Growth in a Turbulent Boundary Layer "* Boundary Layer Meteorol, 2018, 169, 1-10

---

## Author Response (AR3)

The reviewer#2 asked for a rejection of the paper. The main critic was that only 2 simulations were used to fit the model, and the same simulations were reused to estimate its performances.

To answer this, 6 new simulations were run with Meso-NH. This new dataset (called calibration dataset) has been used to calibrate the model, and the two LESs originally used have been used for the validation.

Due to such a major change, the paper have been consequently rebuilt, and we believe it answers all the reasons for rejection given by reviewer #2. A quick answer to them is given in blue in the following lines.

That nearly all choices are informed by comparing to a very limited number of unconverged LES results of an extremely small turbine makes it difficult to judge whether those choices are valid

Appendix of the companion paper showed that the data are converged

The operating conditions used to generate the LES dataset are too similar (CT, TI and tip speed ratio) to be able to extrapolate to any other conditions.

The new calibration dataset has different Ct and TI values

That this small dataset is then also used to judge how well the analytical model performs seems bizarre.

A new calibration dataset has been used so now the validation and calibration are performed on two different datasets

In this regard the authors should try to clarify the aim of the comparison and be more quantitative when judging the accuracy of their new model. From the current presentation it also seems difficult to discern whether there is an advantage from using this more complicated model.

Mentions to new applications that requires atmospheric stability have been added

It has more free parameters and therefore is likely to fit more closely in certain conditions, however it is thus also susceptible to overfitting, calling for using a large LES dataset in its calibration. Large open LES datasets that could be used for this purpose are nowadays widely available.

The authors agree, although the calibration requires the velocity and turbulence In the MFOR, which are usually not included in open datasets.

It would also be interesting to check the connection between stability, meandering, turbine size and boundary layer height and how they influence the envisioned model formulation.

The authors agree but it would require a lot of new LES simulations, which are expensive and time-consuming to run. Such study will hopefully be done in the future.